# Composition and paleoecology of the mid-Jurassic bony fish assemblage of the Vaches Noires cliffs (Normandy, France)

Simon Beaufils[1,2,3‡]*, Romain Vullo[1�], Damien Gendry[1�], Laurent Picot[2�], Lionel Cavin[3�]

1 Université de Rennes, CNRS, Géosciences Rennes, UMR 6118, Rennes, France, 2 Paléospace Muséum, Villers-sur-Mer, France, 3 Department of Geology and Palaeontology, Natural History Museum, Geneva, Switzerland

‡ Principal author on this work.
342 These authors contributed equally to this work.
* beaufils.simon42@gmail.com

## Abstract

The Callovian fossil assemblage of the Vaches Noires cliffs in Normandy, France has played an important role in French paleontology since the end of the 18th century. The bony fish component, which includes coelacanths and ray-finned fishes, is represented mainly by fragmentary material. In this study, we provide the first overview of the bony fish assemblage by examining isolated remains collected mostly by amateur paleontologists over the years and housed in public collections. The assemblage comprises remains of Actinistia (*Trachymetopon* sp.), Pachycormidae are represented by the filter feeder asthenocormin *Leesdichthys problematicus*, and by hypsocormin *Hypsocormus* cf. *leedsi* and *Orthocormus* cf. *tenuirostris*. Ginglymodi are represented by cranial elements of cf. *Scheenstia* and Halecomorphi by '*Eurypoma*' *grande*. Pycnodontiformes (*Mesturus* sp. and *Athrodon* sp.) are represented by isolated vomers and prearticulars. The presence of small-sized actinopterygians is attested by small amphicoelous vertebrae. For each taxon, estimated body size and trophic level are inferred through comparisons with extant ecological analogs. Finally, we compare the Vaches Noires assemblage with other Jurassic fish assemblages to assess broader evolutionary and ecological trends in bony fish diversity during the Jurassic.

## Introduction

The fossil site of the Vaches Noires cliffs and surrounding regions (France, Normandy, Calvados) is important in the history of paleontology and has been studied for over 250 years [1,2]. Public and private collections testify to the intense research carried out in the region since the 18th century. The first references to fossils date back from 18th century, with Abbé Jacques-Francois Dicquemare (1733–1789) who

---

**Data availability statement:** All relevant data are within the manuscript.

**Funding:** This work is a contribution to the project 'Burst and Stasis in morphological evolution of Mesozoic coelacanths' funded by the Swiss National Science Foundation (https://data.snf.ch/grants/grant/207903 to L.C.). The funders had no role in study design, data collection and analysis, decision to publish, or preparation of the manuscript.

**Competing interests:** The authors declare that they have no competing interests.

published an article entirely devoted to the 'osteoliths' from the Kimmeridgian of the Cap de la Hève and the Callovian-Oxfordian series of the Vaches Noires [1,3]. In the 19th century, local collectors gave Georges Cuvier access to fossil material, which allowed him to prove the existence of previously unknown extinct species [1–6]. In more recent times, private collectors have continued to amass extensive fossil assemblages, further enriching the paleontological record through their dedication to the discovery and preservation of significant specimens. Among those who made their collections available to researchers and public institutions are E. and G. Pennetier, M. and J. Charles, D. Gendry and C. Bara [7–9], this paper]. Most of the published fossils from the Vaches Noires cliffs and particularly from the Marnes de Dives Formation are precisely located in the local stratigraphy, corresponding to the Upper Callovian, and the Lamberti Zone. Few fossils not found in situ were removed from the study, to avoid uncertainties in the succession of the ichthyofauna assemblages. Contrary to common belief, fish remains in the Vaches Noires cliffs are abundant, but their fragmentary nature sometimes makes their systematic identification difficult.

During the late Callovian, central England and northern France were covered by an epicontinental sea, as evidenced today by the thick marl series [7,8–11]. This gigantic mudflat open to the offshore waters received terrigenous sediments from the Armorican massif as evidenced by the Marnes de Dives Formation [7,8,9,12]. The driftwood, seeds, and cones found on the rocky platform just above the vertebrates are indicators of terrigenous inputs via storms (oyster-rich limestone deposits and settling of clays with wood and seeds). The Marnes de Dives assemblage includes gastropods, bivalves, cephalopods, particularly ammonites, used for biochronological studies [8,9,12,13]. Vertebrates are abundant, including cartilaginous fishes (mainly hybodontiforms [14]), marine reptiles (plesiosaurs, ichthyosaurs [15] and crocodilians [16]). Some remains of terrestrial animals such as dinosaurs have been rarely found [17–20].

Bony fishes are well represented in the Vaches Noires cliffs since they include ginglymodians (Lepidotidae), pycnodonts [21,22], coelacanths [23–25], representatives of thunniform (Hypsocorminae) and suspension feeder pachycormids (Asthenocorminae) [] 26–31 and halecomorphs [22,32]. Abundant well ossified vertebrae are attributed to undetermined actinopterygians.

The aim of this paper is on the one hand to depict the taxonomic composition and the food chain of bony fishes based on previously described material and new specimens, and on the other hand to integrate it within the whole vertebrate assemblage of the upper Callovian of the Vaches Noires cliffs. Finally, we place and compare the trophic web within the general Mesozoic context of pelagic fish faunas in order to understand the beginnings of the Late Jurassic actinopterygian radiations [33].

## Geological setting

The fossil site of the Vaches Noires cliffs is located in the northwest of Calvados department in France. The cliffs extend over 4.5 km of coastline between the towns of Villers-sur-Mer and Houlgate (Fig 1). The visible sedimentary series outcrops continuously from the upper Callovian (≈163 Ma) to the Middle Oxfordian

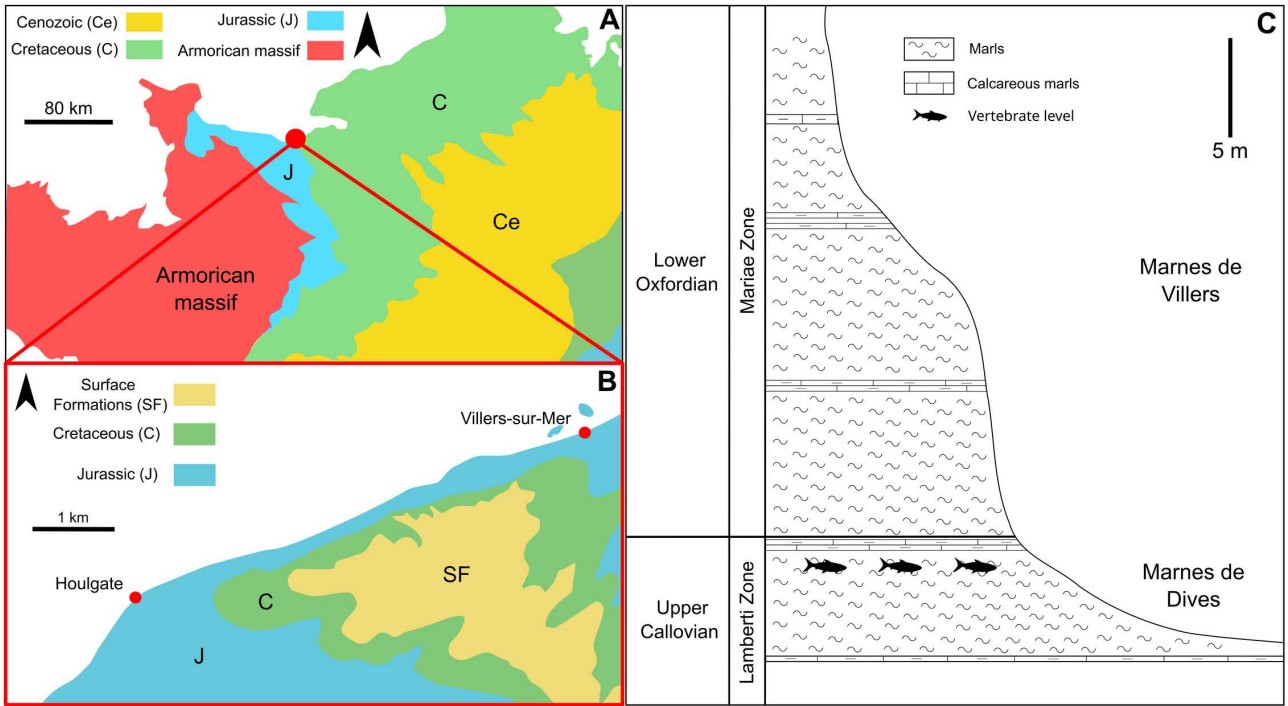

**Fig 1. Geology and stratigraphy of the studied area. A,** geological map of northwestern France showing the location of the Vaches Noires cliffs; **B,** simplified geological map of the site of the Vaches Noires cliffs; **C,** stratigraphic profile of the Vaches Noires cliffs from Marnes de Dives Formation up to Marnes de Villers Formation, black fishes indicate vertebrate-bearing level.

(≈ 160 Ma), the upper part of which is formed by the "coral – rag" (Tenuiserratum Zone) [8,9,34], i.e., an oyster-rich limestone facies deposited in marine subtidal proximal environments. The Albian-Cenomanian flint chalk rests in unconformity above the Jurassic deposits. The Marnes de Dives Formation is well-known for their beds rich in vertebrates dating from the upper Callovian (Lamberti Zone, Praelamberti Subzone). Approximately 15 metres of the 20-metre-thick formation are exposed [13]. These marls are the French equivalent of the British Oxford Clay and outcrop in some areas, such as a rocky flat [7,23,24]. On the Villers-sur-Mer side, the upper Callovian is covered with a recent layer of sand 50 cm to 1 m thick, sometimes uncovered during high tides. Only a few *Leedsichthys* bones recovered from the Henrici Subzone complement the material described from the studied level. All other known specimens originate from the Praelamberti Subzone. The fossiliferous bed is more often exposed on the Houlgate side where several lithostratigraphic markers allow to delimit the different beds between the soft clay-silty layers and the more indurated marl layers [8,9]. Benthic bioclasts are rarely pyritized and contain brachiopods, bivalves such as oysters, gastropods and trigonids all associated with pieces of pyritized wood often decametric in size [8,9]. An assemblage of foraminifera found associated with a vertebrate fossil from the Marnes de Dive provided a Middle Jurassic age, without further precision [24]. Sequential stratigraphy indicates a dynamics of High Stand System Tracts (HSST) at the end of the Callovian [9], with essentially marine influences but also significant terrigenous sedimentary components, including illite, kaolinites and smectites clays [8,9].

## Materials and methods

The list of the material studied (Table 1) includes bony fish remains available for study in the collections of the Paleospace (MPV: Museum Paléospace Villers) and the geological museum of the University of Rennes (IGR: Institut de Géologie

**Table 1. Summary table of the material available for the study of the Callovian ichthyofauna from the Vaches Noires cliffs.**

| Taxa | Bone | Collection | Collection Number |
|---|---|---|---|
| *Trachymetopon* sp. | Pterygoid | Bara | **IGR 153441** |
| *Trachymetopon* sp. | Pterygoid | Bara | **IGR 153442** |
| *Trachymetopon* sp. | Pterygoid | Le Mort | **MPV 2023.2.3** |
| *Trachymetopon* sp. | Palatoquadrate | Gendry | **IGR 153443** |
| *Trachymetopon* sp. | Basisphenoid | Bara – Gendry | **IGR 153444** |
| *Trachymetopon* sp. | Basisphenoid | Bara – Gendry | **IGR 153445** |
| *Trachymetopon* sp. | Basisphenoid | Daumont | **MPV 2024.1.1** |
| *Leedsichthys problematicus* | Nodule | Pennetier | **Batch MPV 2021.1.3.1** |
| *Leedsichthys problematicus* | Ceratobranchial (Halved) | Pennetier | **Batch MPV 2021.1.3.2** |
| *Leedsichthys problematicus* | Ceratobranchial | Pennetier | **Batch MPV 2021.1.3.3** |
| *Leedsichthys problematicus* | Ceratobranchial (two parts) | Pennetier | **Batch MPV 2021.1.3.4** |
| *Leedsichthys problematicus* | Ceratobranchial | Pennetier | **Batch MPV 2021.1.3.5** |
| *Leedsichthys problematicus* | Hypobranchial | Pennetier | **Batch MPV 2021.1.3.6** |
| *Leedsichthys problematicus* | Hyomandibula? | Pennetier | **Batch MPV 2021.1.3.7** |
| *Leedsichthys problematicus* | Ceratobranchial-like element | Gendry | **IGR 153446** |
| *Leedsichthys problematicus* | Cranial element? | Papazian | **MPV 2024.3.2** |
| *Leedsichthys problematicus* | Hypohyal | Daumont | **MPV 2024.1.2** |
| *Leedsichthys problematicus* | Hypohyal | Daumont | **MPV 2024.1.3** |
| *Leedsichthys problematicus* | Hypohyal | Daumont | **MPV 2024.1.4** |
| *Leedsichthys problematicus* | Ceratohyal | Daumont | **MPV 2024.1.5** |
| *Leedsichthys problematicus* | Ceratobranchial-like element | Rivette | **MPV 2024.2.1** |
| *Leedsichthys problematicus* | *Metriorhynchus* stomachal content | Fabrique des Savoirs | **FBS 2012.4.67.30** |
| Hypsocorminae gen. et sp. indet. | Rostrodermethmoid | Le Mort | **MPV 2019.1.1** |
| *Hypsocormus* cf. *leedsi* | Rostrodermethmoid | Le Mort | **MPV 2023.2.1** |
| *Orthocormus* cf. *tenuirostris* | Dentary | Pennetier | **MPV 2022.1.23.1** |
| *Orthocormus* cf. *tenuirostris* | Five coronoid fragments | Pennetier | **MPV 2022.1.23.2-7** |
| Hypsocorminae gen. et sp. indet. | Hypural | Le Mort | **MPV 2023.2.2** |
| Lepidotidae | Basioccipital | Gendry | **IGR 153447** |
| Lepidotidae | Hyomandibula? | Pennetier | **MPV 2022.1.14** |
| Lepidotidae | 6 ganoid scales | Pennetier | **MPV 2022.1.16** |
| Lepidotidae | Jaw fragment | Pennetier | **MPV 2022.1.15** |
| cf. *Scheenstia* | Jaw fragment | Pennetier | **MPV 2022.1.22** |
| 'Eurypoma' grande | Anteroventral skull | Nicolet | **MNHN.F.JRE45** |
| *Athrodon* sp. | Vomer | Drijard | **MPV 2010.6.1** |
| *Athrodon* sp. | Prearticular | Drijard | **MPV 2010.6.2** |
| *Athrodon* sp. | Vomer | Charles | **MPV 2014.2.183** |
| *Mesturus* sp. | Neurocranium | Nicolet | **MNHN.F.JRE46** |
| *Mesturus* sp. | Dermal tessserae | Pennetier | **MPV 2022.1.17** |
| Actinopterygii indet. | 31 vertebrae | Pennetier | **MPV 2022.1.13** |
| Actinopterygii indet. | 33 vertebrae | Charles | **MPV 2014.2.183** |
| Actinopterygii indet. | 47 vertebrae | Ranson | **V176R** |

de Rennes). In addition, we figure bony fishes remains from the National History Museum of Paris (MNHN: Muséum national d'Histoire Naturelle) and a stomach content FBS 2012.4.67.30 associated to a *Metriorhynchus* specimen FBS 2012.4.67.80, housed at la Fabrique des Savoirs (FBS) (Elbeuf, Normandy, France) [35]. All the material likely comes

from the Marnes de Dives Formation and was found either in situ or washed from rocks by the tides. Specimens were prepared by needle when required and MPV 2019.1.1 was prepared by dissolving the matrix with an unknown acid. Specimens were photographed with a camera Olympus TG-6 and drawings based on photographs were made with Inkscape Vector Graphics. Little direct evidence of predator/prey relationship, such as gut contents or traces of predation evidence were available, and consequently the trophic network was reconstructed on indirect evidence such as functional morphology. Trophic levels (TL) are represented by a scale ranging from 1 (primary producers or plants) to 5 (top predators). Pauly and Palomares [36] defined the trophic level of a consumer species i, $TL_i$ as:

$$TL_i = 1 + \sum_j (TL_j \times DC_{ij})$$

Where $TL_j$ is the fractional trophic level of the prey j, and $DC_{ij}$ represents the fraction of j in the diet of i. We cannot directly calculate this value for extinct species. Accordingly, we estimated the TL value of taxa from the Vaches Noires by comparison with TL provided by Fishbase [37] for extant taxa that are phylogenetically and/or ecologically closely related to the extinct taxa. Figures of the fish assemblage network is based on the model proposed by Cavin et al. (2015), Cooper & Martill (2020) et Veiga et al. (2023) [38–40]. The vertebrate trophic chain composition follows the model proposed by Martill et al. (1994) [10]. No permits were required for the described study, which complied with all relevant regulations. Institutional abbreviations as follow: **FBS**: Fabrique des Savoirs; **IGR**: Institut Géologique de Rennes; **MPV**: Muséum Paléospace Villers; **MNHN**: Muséum national d'Histoire Naturelle; **MB**: Museum für Naturkunde der Humboldt-Universität zu Berlin; **NHMUK**: The Natural History Museum, London, U.K.

## Systematic paleontology

Actinopterygii Cope, 1887

Neopterygii Regan, 1923

Holostei Müller, 1846

Halecomorphi Cope, 1872

'*Eurypoma*' *grande* Woodward, 1889

### Material

An anteroventral region of an articulated skull part MNHN.F.JRE45(Nicollet collection).

### Remarks

There is only one mention of halecomorphs at the Vaches Noires Cliffs, that was referred to *Eurycormus grandis* by Wenz [22]. Arratia and Schultze [32] revised the genera *Eurycormus* and *Eurypoma*, and referred the first one to the Teleosteomorpha, more specifically the *Siemensichthys*-group, and the second one to the Amiiformes. Based on this study, the specimen described by Wenz (Fig 2), consisting of a lower part of a skull, was consecutively attributed to the genera *Eurypoma* [32]. Fig 2 shows the specimen and a drawing made after [22,32]. A study of the caturoids of the Vaches Noires cliffs, based on new material, is in progress.

Ginglymodi Cope, 1872

Lepisosteiformes Hay, 1929

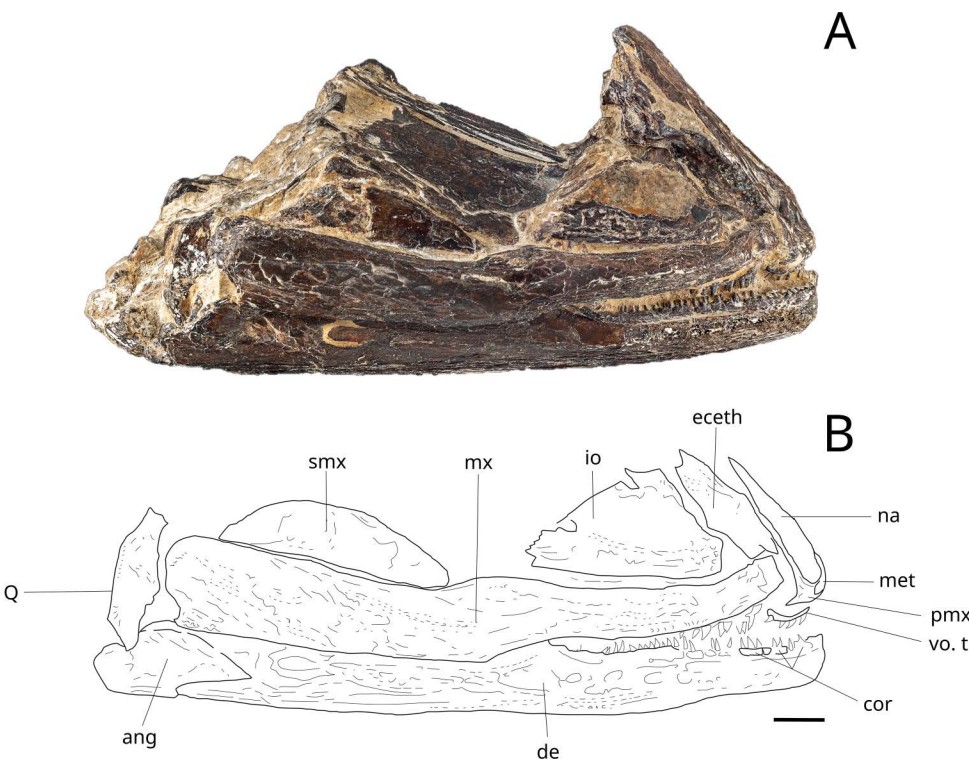

**Fig 2. 'Eurypoma' grande anteroventral skull part MNHN.F.JRE45. (Nicollet collection) Callovian, Villers-sur-Mer, Calvados, right side (Redrawn from Wenz (1968, pl. 38, fig D) and Arratia and Schultze (2007)). ang**, angular; **de**, dentary; **eceth**, ecethmoid; **io**, infraorbital bone (= lacrimal bone); **met**, mesethmoid; **mx**, maxilla; **na**, nasal bone; **pmx**, premaxilla; **q**, quadrate; **smx**, supramaxilla; **vo.t**, vomerine teeth. Scale bar: 10 mm.

Lepidotidae Owen, 1860 (*sensu* López-Arbarello and Wencker, 2016 [41])

Genus and species indet.

## Material

A basioccipital IGR 153447; a hyomandibula MPV 2022.1.14; a batch of 6 ganoid scales MPV 2022.1.16; two jaw fragments MPV 2022.1.15 and MPV 2022.1.22

## Cranial elements

Basioccipital IGR 153447 is 24 mm long (Fig 3A). It was identified by comparison with '*Lepidotes*' *toombsi* (species considered by Schröder et al. (2012) as a semionotiform and included in the genus *Macrosemimimus* [42]) described in detail by Patterson [43]. The occipital condyle incorporates at least one vertebra, as indicated by the pair of facets for ribs present posteroventrally to the condyle. The ventral part shows the sutural surface to the postero-dorsal part of the parasphenoid. The left-wing extension that was sutured to the exoccipital is preserved. The latter forms a 70° angle to the vertical axis of the bone.

The hyomandibula MPV 2022.1.14 (Figs 3B and C) is identified by comparison with the hyomandibula NHMUK PV P.6841(3) of *Lepidotes latifrons* Woodward, 1893 from the NHMUK collections and a '*L.*' *gloriae* hyomandibula USNM 279856 described by Thies [44]. MPV 2022.1.14 measures 53 mm in length and 55 mm in maximum height.

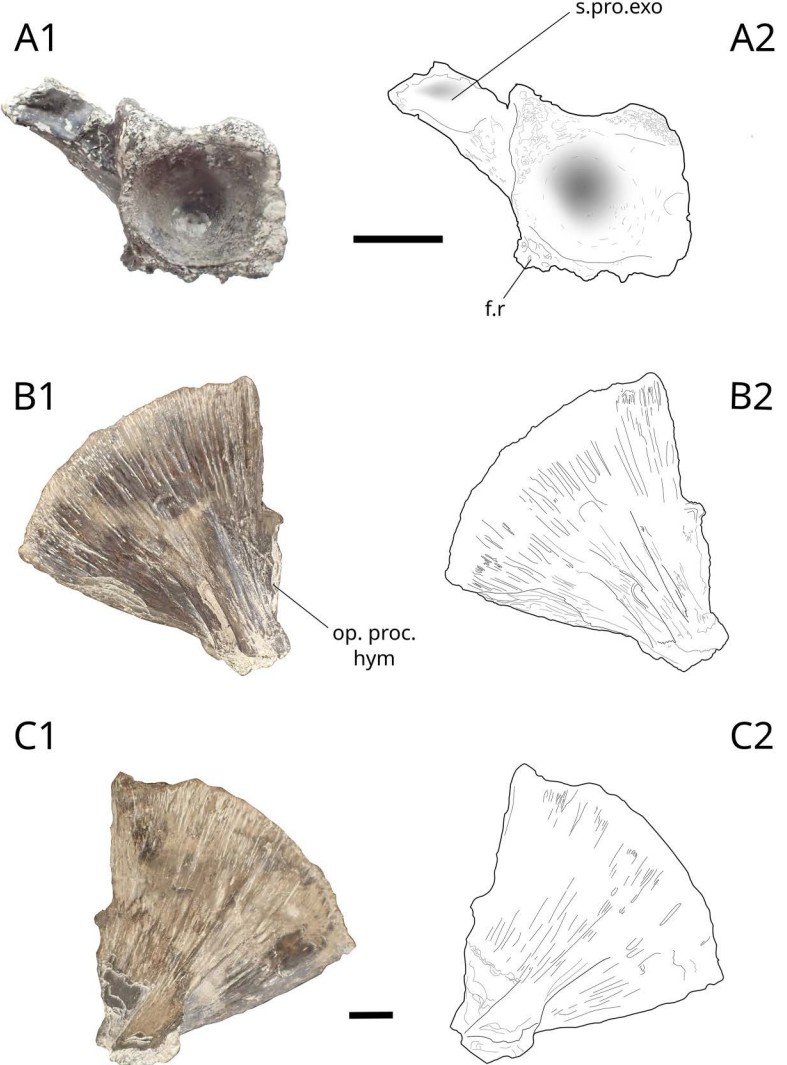

**Fig 3. Lepidotidae indet. cranial elements.** Basioccipital (A) IGR 153447 and hyomandibula (B and C) MPV 2022.1.14. Photograph (A1) and drawing (A2) of the basioccipital IGR 153447 in posterior view. Photograph (B1, C1) and drawing (B2, C2) of left hyomandibula MPV 2022.1.14 in internal view (B) and lateral view (C). **op. proc. hym**, opercular process of the hyomandibula; **f.r**, facet for ribs **s.pro.exo**, suture process with exoccipital. Scales bars: 10 mm.

The anterior lower part is developed in a massive condyle that articulated with the lower jaw. The bone is textured with fine striations radiating from the condyle to the curved upper margin. The lateral surface bears a wing-like extension that forms a shelf and a semi-cylindrical groove. The medial surface bears a shallow crest with a semi-cylindrical channel.

## Ganoid scales

Six scales that have a typical "peg and socket" articulation ([Fig 4]). Following Schultze (2018) [45], ganoine is restricted to the free field. The posterior margin of ganoin cover has a finely fringed edge.

*Scheenstia* López-Arbarello & Sferco, 2011

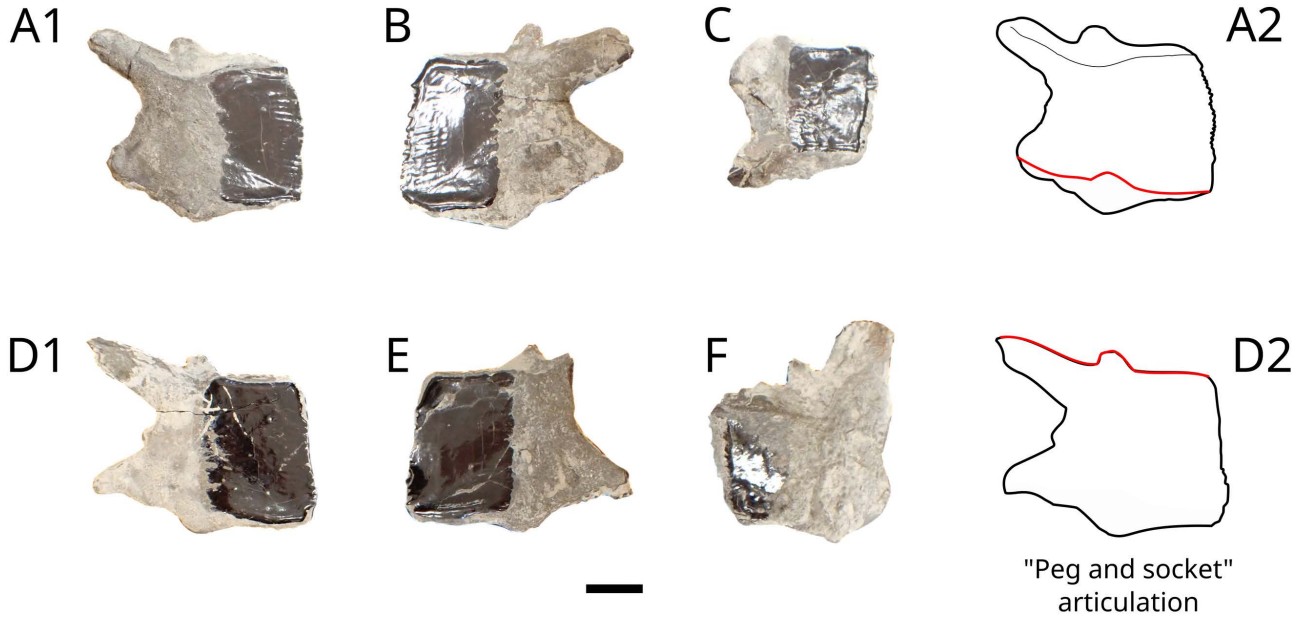

**Fig 4. Lepidotidae indet. ganoid scales MPV 2022.1.16.** Photograph (A1, B, C, D1, E, F) and drawings (A2 and D2) to show the peg and socket articulation. Scale bar: 10 mm.

cf. *Scheenstia* sp.

## Material

A dentary MPV 2022.1.15; an undetermined dentigerous element MPV 2022.1.22.

## Jaw fragments

The dentary (Fig 5A), 150 mm long and 40 mm high, is identified by comparison with material described in [46] and [47]. The ventral margin is concave, and the occlusal surface is convex. A series of openings for the mandibular sensory canal runs in the mid-depth of the dentary along on the anterior half of the bone. There are four complete teeth remaining, three broken and two alveoli. Teeth are styliform and relatively high with a maximum height of 6.1 mm and a maximum diameter of 1.5 mm at the base. The attachment of teeth to the jaw is Type 1 (*sensu* Fink, 1891 [48]) characterized by complete ankylosis of the tooth to the jaw-bone, indicating a crushing mode of feeding.

The undetermined dentigerous fragment is 27 mm long and 11 mm high (Fig 5B). Eight teeth are visible, with two complete ones. They have a very typical acrodine cap at the top, resembling those of *Scheenstia*. The rest of the damaged teeth consists of dentine and a pulp cavity in the centre.

*incertae sedis*

Pycnodontiformes Berg, 1937

## Remarks

At the Vaches Noires Cliffs, no articulated pycnodont skeleton has ever been found. Two families have been recognized on the basis of isolated elements, the pycnodont *incertae sedis Athrodon* sp. [21], and the Mesturidae with *Mesturus* [22].

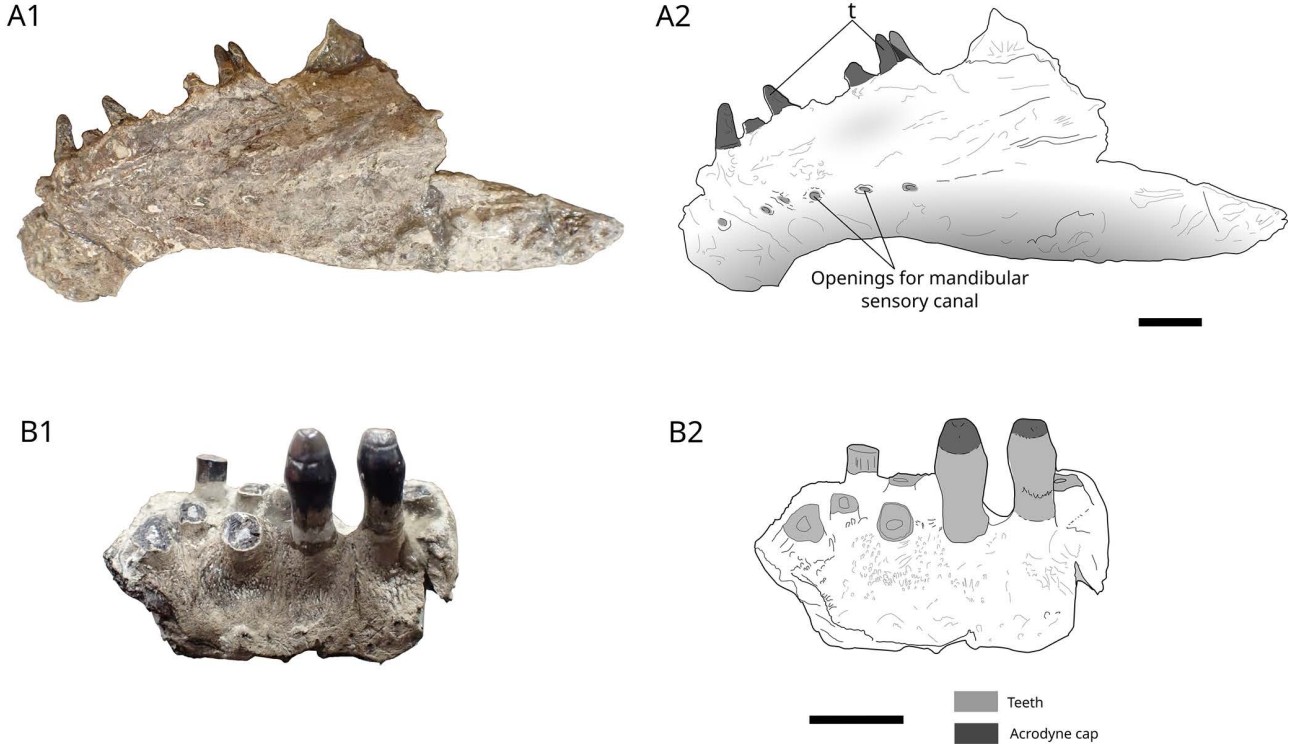

**Fig 5. cf. _Scheenstia_ sp. MPV 2022.1.15 and MPV 2022.1.22.** Photographs (A1, B1) and drawings (A2, B2) of dentary MPV 2022.1.15 in left lateral view and MPV 2022.1.22 dentary fragment in labial view. t, teeth. Scales bars: 10 mm.

Fragments of vomers, prearticular, premaxillary and isolated teeth have been regularly found in the upper Callovian of the Vaches Noires Cliffs. From this level two almost complete vomer (MPV 2014.2.183 and MPV 2010.6.1), a prearticular fragment (MPV 2010.6.2), an endocranium (MHNH.F.JRE46, identification from [22]) from the Nicollet collection and a dermal tesserae from the Pennetier collection in Paleospace are studied.

_incertae sedis_

_Athrodon_ Sauvage, 1880

_Athrodon_ sp.

## Material

Two vomer MPV 2010.6.1 and MPV 2014.2.183; two prearticulars MPV 2010.6.2 and MB f. 1337.

## Vomer

The first vomer MPV 2010.6.1 (Fig 6 A1 and A2) has a maximum length of 19 mm long and is the most complete of the two vomers. There are seven rows with six teeth preserved on the Vomer Medial Row (V.M.R.). There are three rows per side with one V.P.L.R (Vomer Primary Lateral Row), one V.S.L.R (Vomer Second Lateral Row) and one V.T.L.R (Vomer Tertiary Lateral Row). The second vomer MPV 2014.2.183 (Fig 6 B1 and B2) has a maximum length of 17 mm long and maximum width of 10 mm. There are at least five rows with four teeth preserved on the Vomer Medial Row (V.M.R.). There

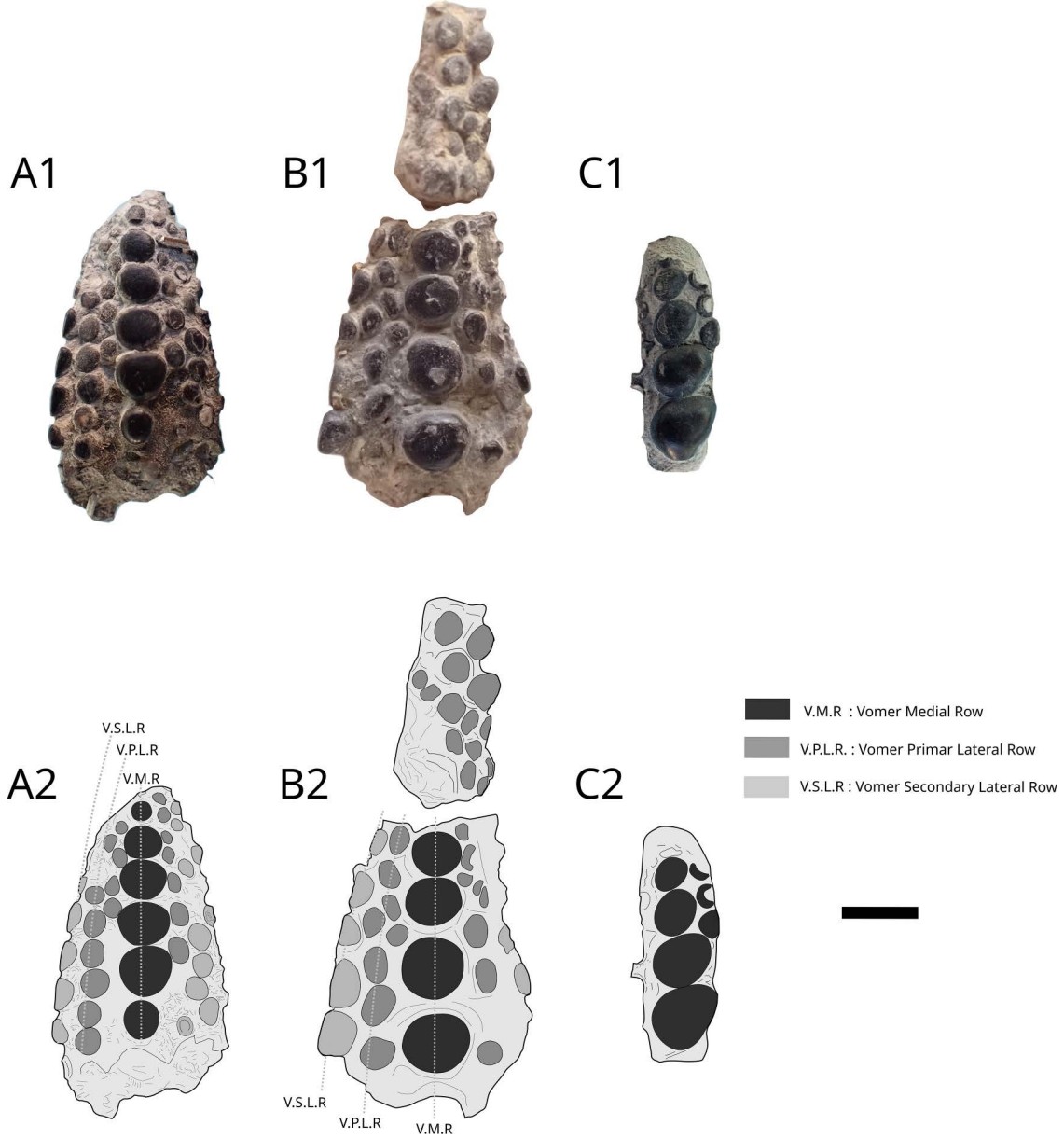

**Fig 6. *Athrodon* sp. vomers MPV 2010.6.1, MPV 2014.2.183 and prearticular MPV 2010.6.2.** Photographs (up) and drawings (down). Vomer from display cases at Paleospace (A1, A2); MPV 2014.2.183 (B1, B2); prearticular fragment from display cases at Paleospace museum (C1, C2). Scale bar: 5 mm.

are three rows on the left side with one V.P.L.R, one V.S.L.R and one V.T.L.R and two rows on the right side with one V.P.L.R and one V.S.L.R.

A fragment of prearticular MPV 2010.6.2 (Fig 6C1 and C2) bearing four large teeth and three small ones, which are rather similar but less symmetrical than teeth from the vomers is present. We attribute these fragments to *Athrodon* sp. because the teeth are smooth, and uncrenulated unlike *Mesturus* and *Gyrodus*, which both exhibits strongly crenulated tooth. We also rule out *Eomesodon* and *Proscinetes* by comparison with Licht [49]. Both genera possess vomerine and

prearticular teeth with an elliptical shape, which differs from the more rounded, circular teeth observed in all specimens. However, vomerines tooth rows are less numerous than in the vomer of *Athrodon* MB f. 7135 assigned to *A. wittei* with at least four lateral rows on each side of the V.M.R. as described by Kriwet (23). Moreover, the presence of *Athrodon* has been confirmed with the prearticular MB f. 1337 from the Vaches Noires cliffs described by Kriwet [21].

Mesturidae Nursall, 1996

*Mesturus* Wagner, 1862

*Mesturus* sp.

## Material

Endocranium MNHN.F.JRE46 (Nicollet collection); Dermal Tesserae MPV 2022.1.17.

## Endocranium

A part of an endocranium referred as a large *Mesturus* skull described by Wenz [22] is redrawn here (Fig 7) and a fragment of dermal tesserae figured in Fig 8. The endocranium was described for the first time by Wenz [22]. The attribution to *Mesturus* is supported by the comparison between the ornamentation of the dermal bones and the dermal tesserae with the squamation pattern observed in *Mesturus leedsi,* Woodward, 1895, which shows a good match [50]. Moreover, the specimen falls within the appropriate temporal range for this genus (Callovian to Tithonian) [51].

Teleosteomorpha Arratia, 1999

Pachycormiformes Berg, 1937

Pachycormidae Woodward, 1895

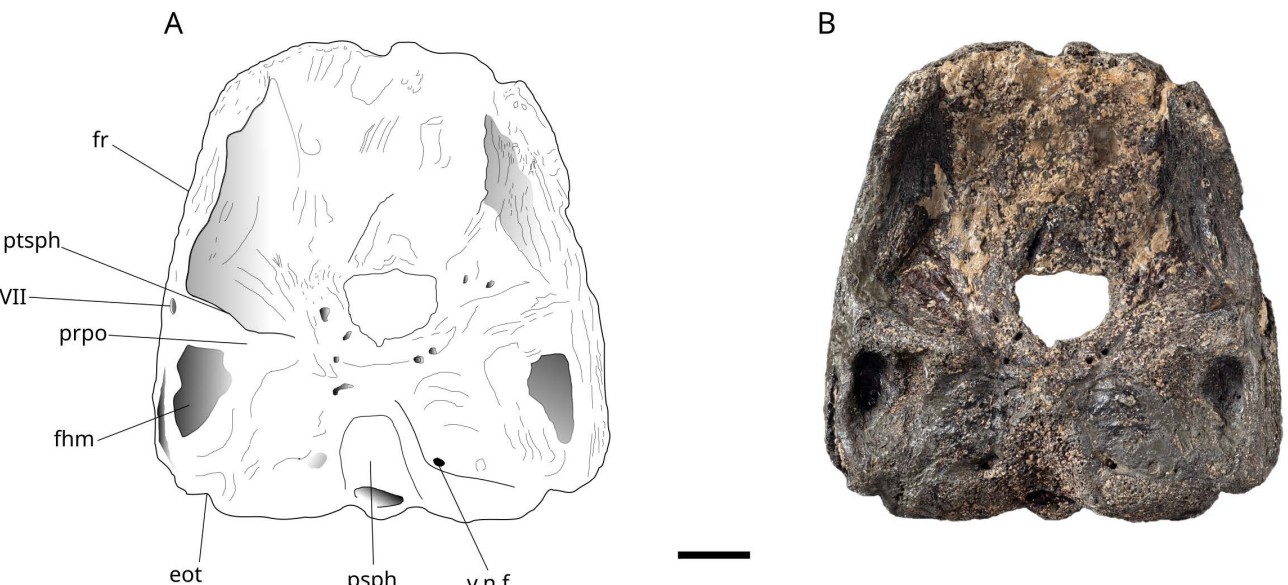

**Fig 7. *Mesturus* sp. MNHN.F.JRE46 endocranium in ventral view. eot**, Epiotic; **fhm**, Facet for articulation of the Hyomandibula; **fr**, Frontal; **prpo**, Postorbital process; **pSph**, Parasphenoid; **ptsph**, Pterosphenoid; **v.n.f**, Vagus nerv foramen; **VII**, Facial nerve foramen. Scale bar: 20 mm.

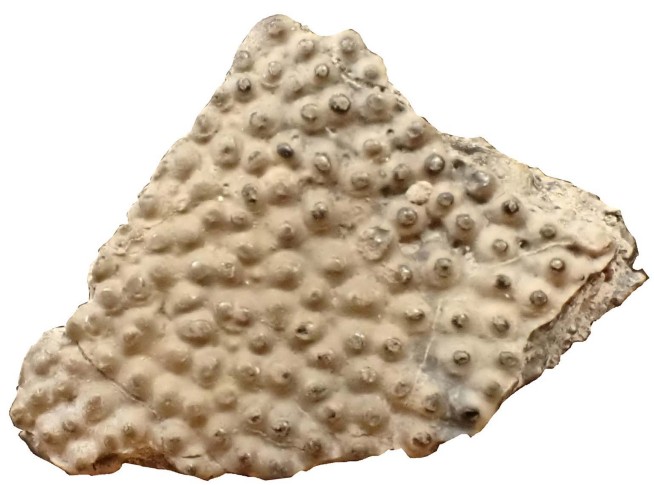

**Fig 8. *Mesturus* sp. MPV 2022.1.17 fragment of dermal tesserae.** Scale bar: 10 mm.

## Remarks

The Pachycormidae are divided into two subfamilies: Hypsocorminae and Asthenocorminae [52], the latter comprising suspension feeding Pachycormidae. Recent phylogenetic analyses indicate that pachycormids are closely related to stem Teleostei, grouping them together within Teleosteomorpha. [53–56].

Asthenocorminae Cooper, Giles, Young, Maxwell, 2022

*Leedsichthys* Woodward, 1889

*Leedsichthys problematicus* Woodward, 1889

## Material

Nine specimens referred to *L. problematicus* MPV 2021.1.3 (Paleospace collections) (One nodule MPV 2021.1.3.1, four ceratobranchial parts MPV 2021.1.3.2 to 5 (with 2021.1.3.3 identified by Liston [27]), one hyomandibula? MPV 2021.1.3.6, one hypobranchial MPV 2021.1.3.7); one part of a dorsal fin or ceratobranchial IGR 153446 (Gendry collection); an undetermined opercular or cranial bone MPV 2024.3.2; ceratohyal MPV 2024.1.5; ceratobranchial-like element MPV 2024.2.1; three hypohyals MPV 2024.1.2, MPV 2024.1.3 and MPV 2024.1.4; stomachal content from a *Metriorhynchus* FBS 2012.4.67.30 (La Fabrique des Savoirs collections).

## Remarks

For over 100 years, remains of *L. problematicus* have been found in the British Oxford Clay [3,27,57], the Vaches Noires cliffs [26], and also more recently in Chile (Atacama Desert) [26,58]. *Leedsichthys problematicus*, Woodward, 1889 [59] belongs to the Asthenocorminae, with other genera such as the Toarcian *Pachycormus* Agassiz, 1833 [60], *Martillichthys* Liston, 2008 (Upper Callovian), *Asthenocormus* Woodward, 1895 (Tithonian), *Bonnerichthys* Friedman et al., 2010 [61] and *Rhinconichthys* Friedman et al., 2010 [61] (Upper Cretaceous). Bones from this species have long been confused with parts of thyreophoran dinosaurs, including *Lexovisaurus* (parietals confused with parts of armor plates, and

ceratobranchial confused with caudal spines) [27,62]. There is currently only one species of *Leedsichthys*, *L. problematicus* [27]. In Normandy, the genus *Leedsichthys* is reported from the late Bathonian (Bavent quarry) [63] to the Kimmeridgian (Cap de la Hève, Seine-Maritime) [27].

### Gill rakers

MPV 2021.1.3.1 is a nodule that measures 123 mm by 105 mm containing gill rakers of *L. problematicus* (Fig 9) (MPV 2021.1.3), as well as a tooth of the shark *Notidanoides* and on the back of the nodule, one ammonite *Kosmoceras spinosum.* The latter defines the Praelamberti Subzone in the Lamberti Zone. The nodule contains four fragments of gill rakers. Gill rakers 1 and 2 are the best preserved, measuring 62 mm and 60 mm maximum respectively. They consist of a base, a branch and fimbriations, fringes are inserted between the dental needles. FBS 2012.4.67.30 (Fig 10), corresponds to a stomachal content from a *Metriorhynchus* (Thalattosuchia) with two gill rakers, 37 mm and 58 mm in length. Gill rakers 3 and 4 are poorly preserved. As observed in FBS 2012.4.67.30 and MPV 2021.1.3, there is interfanuncular mesh fragments *sensu* Liston (2013a) and interfanuncular *sensu* Liston [26,64,65]. Structures like this linked gill rakers together to form a functional sieve [64]. This specimen has been recently described in a study on the diet of *Metriorhynchus* [35].

### Ceratobranchials

Ceratobranchial specimens MPV 2021.1.3.2 to 2021.1.3.5 are illustrated in Fig 11A–E, including a fragment cut in half used for making thin sections figured by Liston [27] (Fig 11A1 and A2). A second halved ceratobranchial fragment was analyzed to assess its uncrushed triangular shape. Several additional fragments are attributed to *L. problematicus* ceratobranchials based on their characteristic triangular cross-section. These bones served as attachment sites for gill rakers. Early Callovian occurrences have been documented by Liston & Gendry (2015) [63].

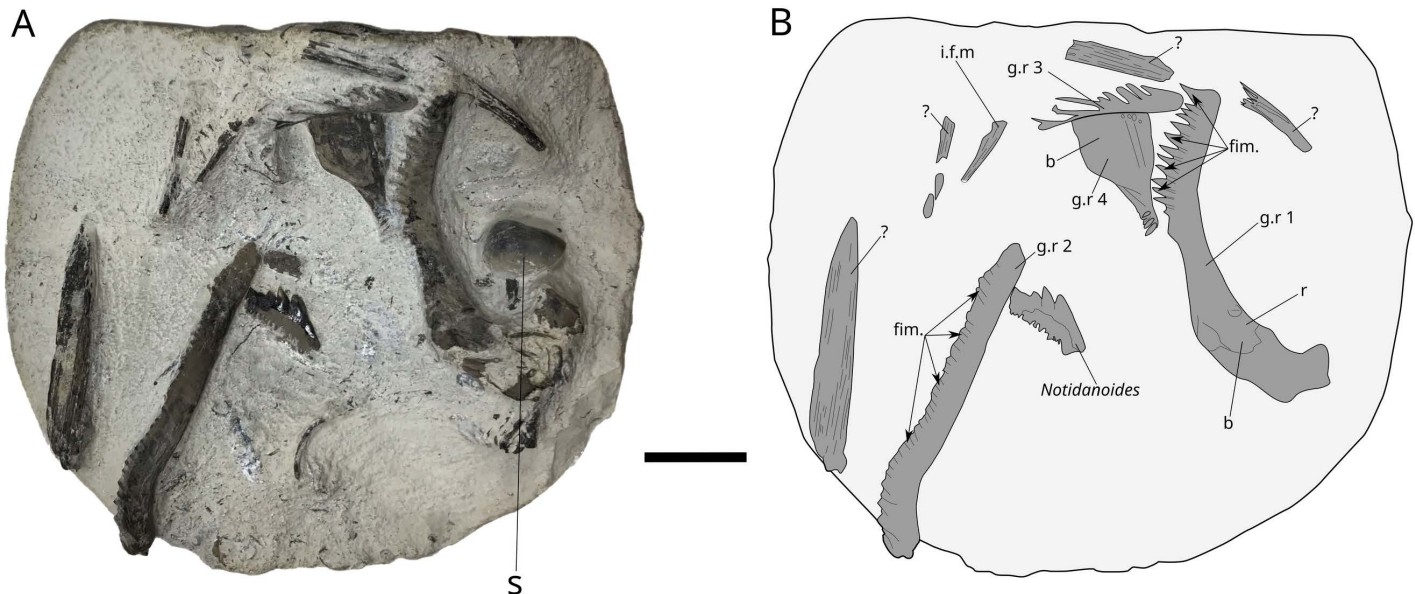

**Fig 9. *L. problematicus*. MPV 2021.1.3.1. photograph (A) and drawing (B) of the gill rakers nodule. b**, base; **fim.**, fimbriation; **g.r.**, gill raker; **i.f.m**, interfanuncular mesh fragment; **s**, shell fragment?, undetermined element. Scale bar: 20 mm.

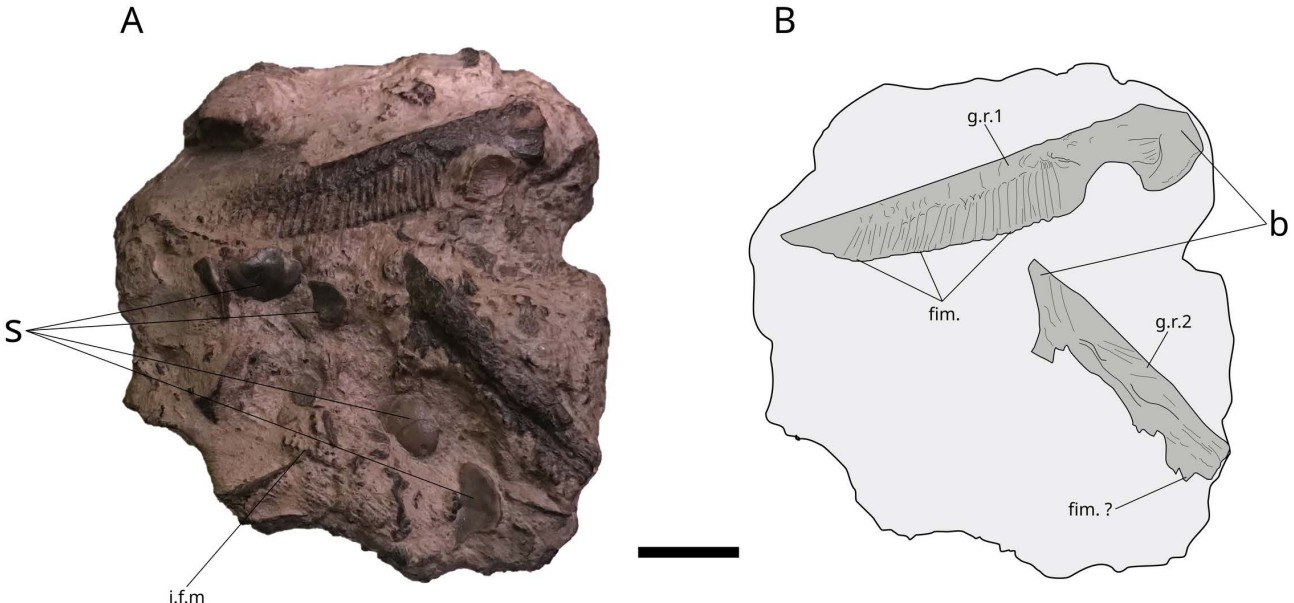

**Fig 10. Metriorhynchidae stomachal content FBS 2012.4.67.30 (Fabrique des Savoirs).** Photograph (A) and drawing (B) of stomachal content nodule of a thalattosuchian (Metriorhynchidae) containing gills rakers of L. problematicus. **b**, base; **fim**, fimbriations; **g.r.**, gill rakers; **i.f.m**, interfanuncular mesh fragment; **s**, shell fragment. Scale bar: 20 mm.

## Hypobranchial

A fragment of a right hypobranchial MPV 2021.1.3.6 (Fig 11E) from the anterior part of the gill basket (identified by comparison with NHMUK P. 10156 [27]). This 246 mm long bone is well ossified. The anterior part, more massive, is relatively well preserved.

## Skull roof elements?

Probable cranial elements of *L. problematicus.* MPV 2021.1.3.7 (Fig 11F) and MPV 2024.3.2 (Fig 12). MPV 2021.1.3.7 is 145 mm in length, but more fragmentary than MPV 2024.3.2 which is 365 mm in length. The two bones have a central hump often observed in dermal bones of *L. problematicus* [27,66]. In each bone, striations radiate from the ossification center to the external peripheral shelf that is blade-like. This feature is more clearly visible in MPV 2024.3.2, which is significantly more complete, whereas in MPV 2021.1.3.7, the borders are fragmented.

## Hypohyals

Three hypohyals (Fig 13), one right MPV 2024.1.2 (Fig 13A1, B1), two left MPV 2024.1.3 (Fig 13A2, B2) and MPV 2024.1.4 (Fig 13A3, B3). They were identified by comparison with hypohyals of *Leedsichthys* described by Liston [66]. These bones are well-ossified. MPV 2024.1.2 is 190 mm in length, MPV 2024.1.3 is 182 mm in length, MPV 2024.1.4 is 113 mm in length. The ossification state of MPV 2024.1.2 is stronger than MPV 2024.1.3. MPV 2024.1.4 is broken but preserves the anterior part. The dorsal surface of all three specimens is highly ridged anteriorly, with a smooth ventral surface. Three characteristic medial indentations give rise to a shelf that extends antero-ventrally along part of the medial edge.

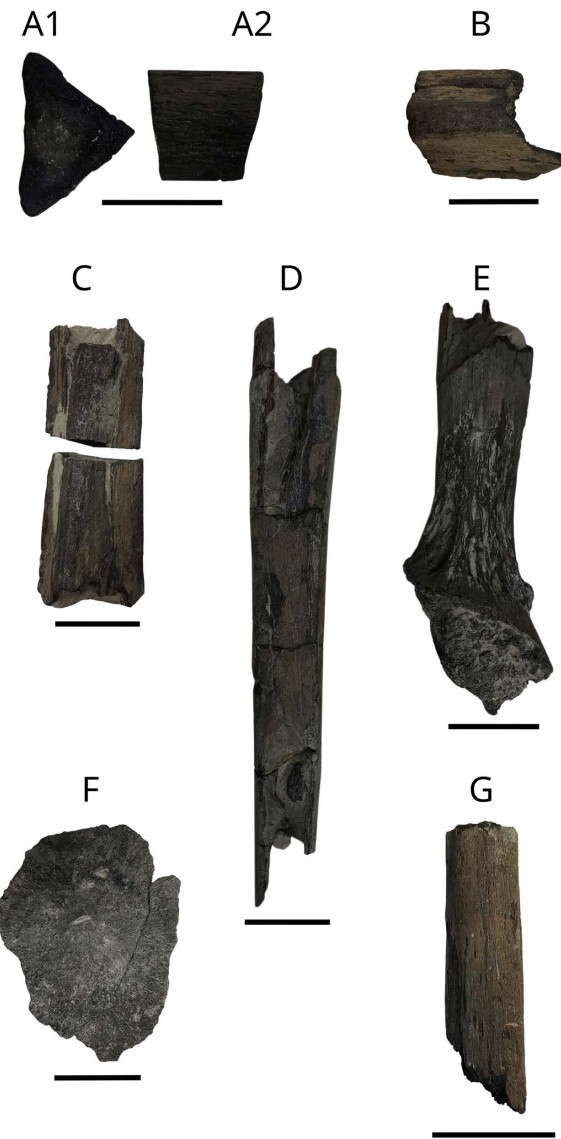

**Fig 11. *L. problematicus* MPV 2021.1.3 batch and ceratobranchial fragment IGR 153446. A, MPV 2021.1.3.2.** Ceratobranchial in transverse (A1) and lateral (A2) views (identified by Liston [27]); B. MPV 2021.1.3.3. Ceratobranchial in lateral view (identified by Liston [28]); C, MPV 2021.1.3.4. Ceratobranchial in lateral view (two parts); D, MPV 2021.1.3.5. Ceratobranchial in lateral view (two parts); E, MPV 2021.1.3.8. Hypobranchial in lateral view; F, MPV 2021.1.3.7. Hyomandibula fragment?; G, IGR 153446 Ceratobranchial?-like element. Scale bar: 5 cm.

## Ceratohyal

An anterior ceratohyal (Fig 14) MPV 2024.1.5 identified based on its typical centrally narrowed shape and in comparison with material described by Liston [27,29,66]. MPV 2024.1.5 is 330 mm maximum length but broken at both anterior and posterior ends. It has a minimum width of 110 mm. The thickest border is the superior part of the bone, the inferior blade-like part runs longitudinally. The bone is rugose on both surfaces, due to the mesh-like ossification-type typical of *Leedsichthys*. A pattern of striations radiates from the center of the convexly curved outer surface. The internal surface is flat. The bone is teardrop-shaped in transverse view.

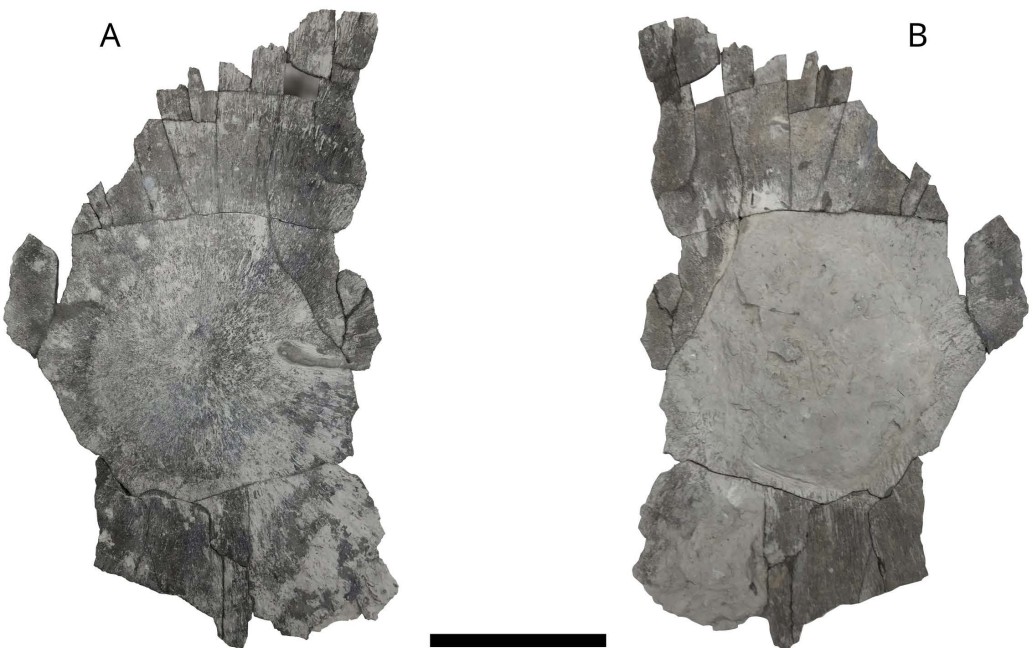

**Fig 12. *L. problematicus* skull roof elements? MPV 2024.3.2.** A, in dorsal view; B, in ventral view. Scale bar: 10 cm.

### Ceratobranchial?

Two ceratobranchial-like fragments (Fig 15), MPV 2024.2.1 and IGR 153446 (Fig 11G), have been identified as belonging to *Leedsichthys problematicus* based on comparative morphology with previously described material, particularly specimens SMC J46873 for MPV 2024.2.1 [29] and NHMUK P.6925 and NHMUK P.6928 [27,28,66] for IGR 153446. The fragment MPV 2024.2.1, which measures 440 mm in length, is conical, anteroposteriorly compressed, and exhibits a slight curvature. In contrast, IGR 153446 is notably shorter, 114 mm, and has a more circular cross-section. Both fragments display a rugose surface texture and a distinctive mesh-like ossification-type typical of *Leedsichthys*.

Hypsocorminae Vetter 1881 *sensu* Cooper et al., 2022

### Remarks

The second pachycormid clade present in the assemblage is the one composed exclusively of macrocarnivorous fishes. This clade contains thunniform species, such as the Middle/Late Jurassic *Hypsocormus* Wagner, 1863 or the Late Cretaceous swordfish-like *Protosphyraena* Leidy, 1857 [67]. The rostrodermethmoid is a specific bone resulting either from the fusion of ethmoid and vomer [68–71], or from the fusion of rostral and dermethmoid [55].

### Material

Rostrodermethmoids MPV 2019.1.1 and 2023.2.1; Batch MPV 2022.1.23 comprising dentary part (MPV 2022.1.23.1), 5 coronoids fragments (MPV 2022.1.23.2 to 2022.1.23.5) (and one tooth not figured here MPV 2022.1.23.6) and one coronoid fragment (MPV 2022.1.23.7); One hypural bone MPV 2023.2.2.

Hypsocorminae gen. et sp. indet.

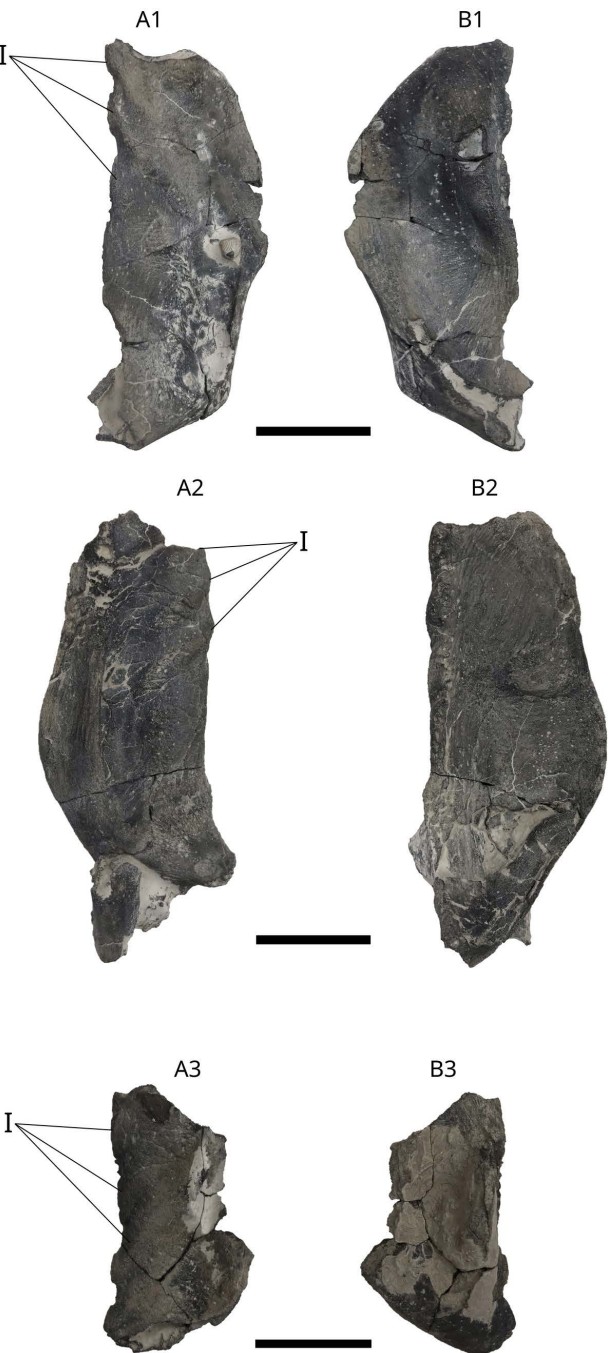

**Fig 13. *L. problematicus* right hypohyal MPV 2024.1.2, left hypohyal MPV 2024.1.3 and left hypohyal MPV 2024.1.4.** MPV 2024.1.2 A1, in dorsal view; B1, in ventral view. MPV 2024.1.3 A2, in dorsal view; B2, in ventral view. MPV 2024.1.4 A3, in dorsal view; B3, in ventral view. **I**, Indentations. Scale bar: 5 cm.

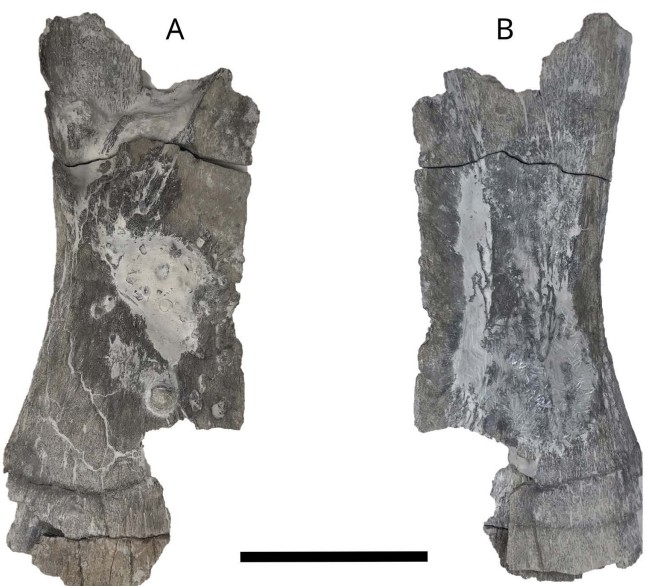

**Fig 14. *L. problematicus* right ceratohyal MPV 2024.1.5.** A, in lateral view; B, in medial view. Scale bar: 10 cm.

### Rostrodermethmoid

Specimen MPV 2019.1.1 (Fig 16) is identified as a rostrodermethmoid by comparison with NHMUK PV P.6916 (3) and NHMUK PV P.6913. The specimen measures 102 mm long and 71 mm wide maximum. A series of large teeth, of which only the tooth base or the corresponding empty sockets are preserved, extends on the lateral edge of the ossification. Traces of eight of these large teeth are visible on the right side and six on the left. They are arranged irregularly, with some teeth located offset mesially than the main alignment. The space left labially is occupied by smaller teeth, of variable size and irregularly arranged.

A bony fragment located in the matrix that fills the specimen may be part of the left antorbital, as seen in a more complete three-dimensional specimen of *Pachycormus* from the Toarcian [60]. We attribute the rostrodermethmoid to Hypsocorminae gen. and sp. indet. because of the peculiar dentition composed of very robust but peculiarly arranged teeth. *Martillichthys* and *L. problematicus* are dismissed because of their edentulous nature.

### Hypural bone

An almost complete compound hypural plate MPV 2023.2.2 (Fig 17), 18 mm long and 19 mm high at maximum, is identified by comparison with the hypural plate attributed to *Hypsocormus posterodorsalis* in Maxwell et al., 2020 and Maxwell et al., 2025 [72,73]. This plate-like triangular bone formed the central part of a caudal fin. MPV 2022.2.2 is identified as an indeterminate Pachycormidae. Fig 17 shows the groove for what we interpret as the *arteria pinnalis* (Fig 17B) [74] which runs on the upper front part of the hypural bone. The parhypural foramen opens at the level of fusion of the parhypural centrum with the compound hypural plate.

*Hypsocormus* Wagner 1863

*Hypsocormus* cf. *leedsi* Woodward, 1889

A

B

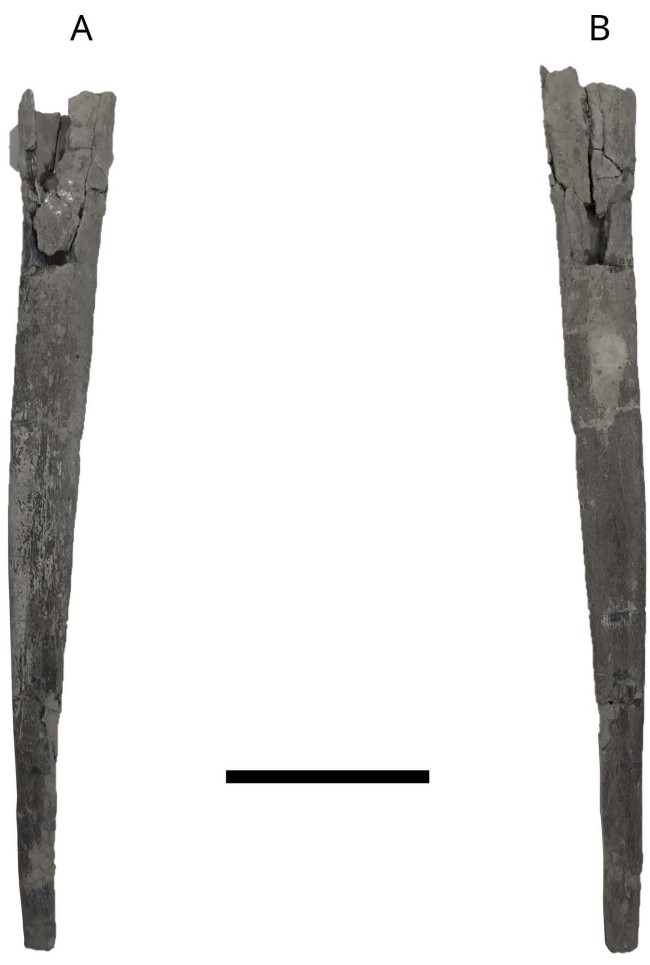

**Fig 15. *L. problematicus* ceratobranchial? MPV 2024.2.1.** A, in lateral view; B, in medial view. Scale bar: 10 cm.

## Rostrodermethmoid

Specimen MPV 2023.2.1 (Fig 18) is identified as a rostrodermethmoid by comparison with NHMUK PV P.6916 (3) and NHMUK PV P.6913. This smaller rostrodermethmoid measures a maximum of 40 mm in length and 38 mm in width (Fig 18). Two proportionally large paramedial teeth profoundly anchored in bone are visible on both sides of the symmetry axis (Fig 18A1 and A2). There is a well-developed symphyseal tooth at the front, anteriorly oriented. Numerous small, staggered lateral marginal teeth are present, primarily along the right labial margin. The dorsal side bears an ornamentation made of small wrinkles (Fig 18B). We identify the specimen as *Hypsocormus* cf. *leedsi*, as it partly matches the NHMUK PV P.6913 specimen, but not fully, particularly with the medial tooth on MPV 2023.2.1. This fragment may represent a new species, but further material is required to test this hypothesis.

*Orthocormus* Weitzel 1930

*Orthocormus* cf. *tenuirostris* [Woodward, 1889]

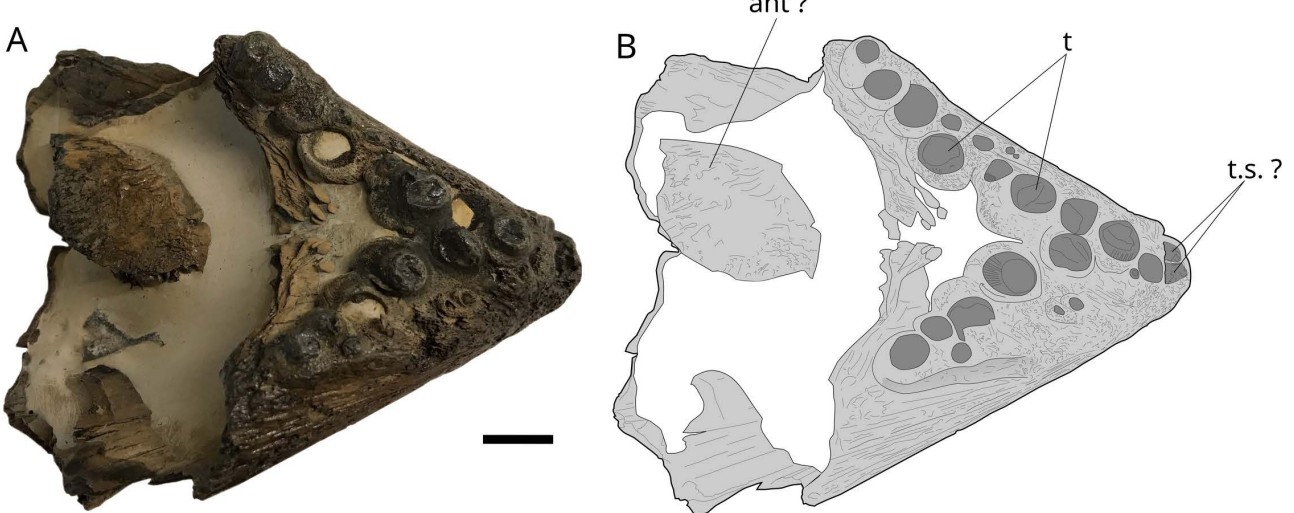

**Fig 16. Hypsocorminae gen. et sp. indet. rostrodermethmoid MPV 2019.** Photograph (A) and drawing (B) of the cf. *Hypsocormus* rostrodermethmoid in ventral view. **ant**, antorbital; **t**, teeth; **t.s.**, teeth sockets. Scale bar: 10 mm.

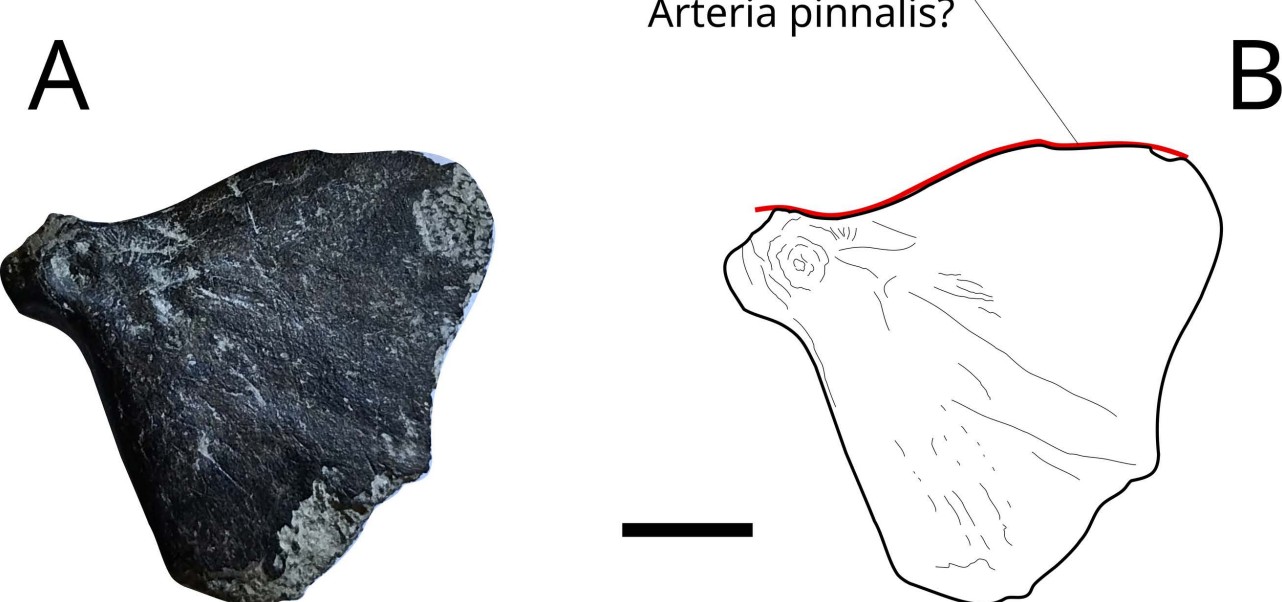

**Fig 17. Hypsocorminae gen. et sp. indet. hypural plate MPV 2023.2.2.** In red, path of the *arteria pinnalis*. Scale bar: 5 mm.

### Jaw fragments

Six fragments are available from MPV 2022.1.23.1 to 5 and MPV 2022.1.23.7 (Figs 19–21). We attribute them to *Orthocormus* cf. *tenuirostris* by comparison with NHMUK PV P.6916 [59] and by comparison with Cooper & Maxwell, 2025 [75]. Discovered close to each other, they probably belong to the same individual. MPV 2022.1.23.1 (Fig 19), identified as

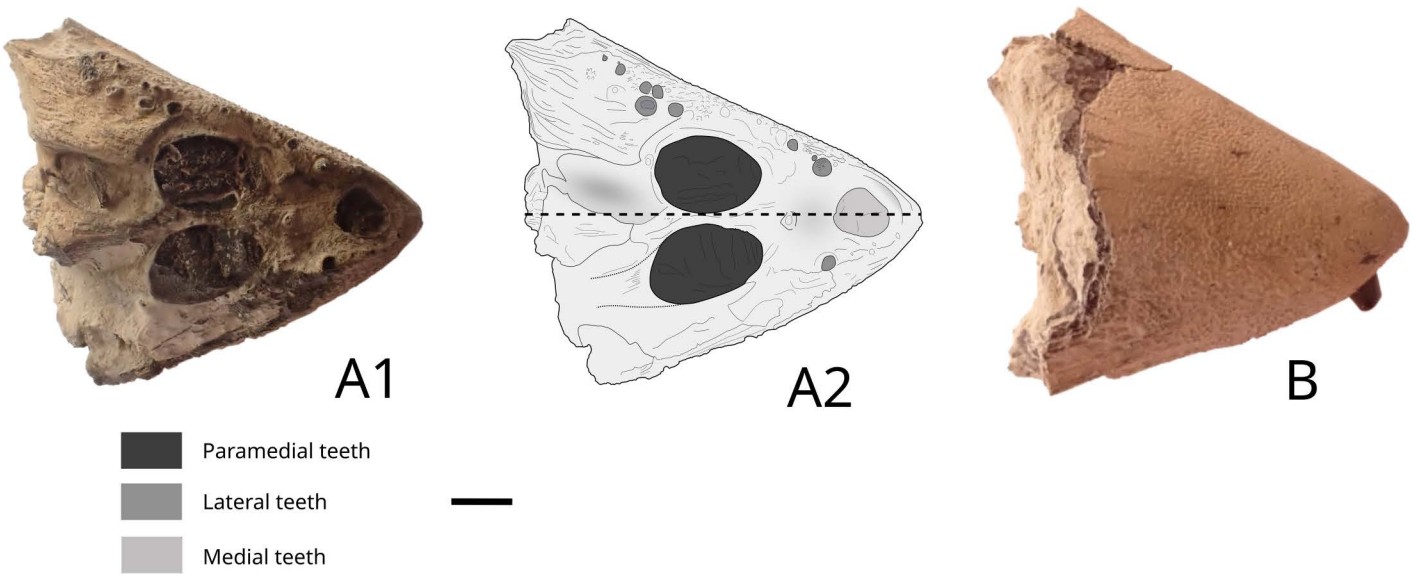

**Fig 18. Hypsocormus cf. leedsi rostrodermethmoid MPV 2023.2.1: photographs (A1 and B) and drawing (A2) of the cf.** *Hypsocormus* cf. *leedsi* rostrodermethmoid in ventral view (A1, A2) and dorsal view (B). Scale bar: 5 mm.

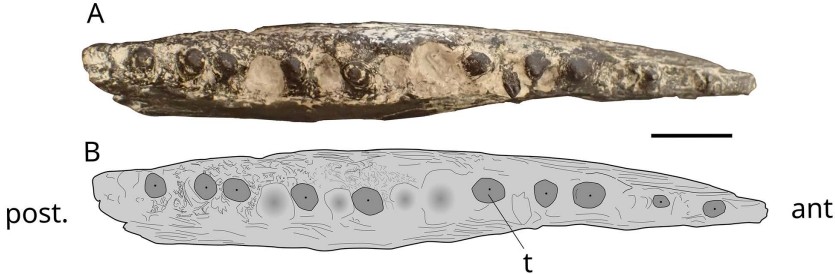

**Fig 19.** *Orthocormus* cf. *tenuirostris* **dentary MPV 2022.1.23.1. Photograph (A) and drawing (B) of a** *Orthocormus* cf. *tenuirostris* **dentary in dorsal view. t**, teeth sockets. Scale bar: 10 mm.

a fragment of dentary [70], is 73 mm long and 9 mm wide at its widest level. It is probably a left dentary, given the curvature of the bone and the way it refines in dorsal view.

12 teeth and seven empty sockets are visible. Teeth seem to be almost fused with the bone. They are well spaced, more in the posterior part than in the anterior part of the ossification [70]. Five other fragments MPV 2022.1.23.2 to 5 (Fig 20) and MPV 2022.1.23.7 (Fig 21) are identified as pieces of coronoids that still bear teeth. Small, 2–4 mm high marginal teeth are visible on the labial edge of some of the coronoid fragments (Fig 20B, C and D and Fig 21). The main teeth are large, thick, and their crown is 10 mm to 20 mm high. In apical view, they have a sub-oval shape, and their base is funnel-like. The surface of all teeth is striated longitudinally. The most distal part of teeth seems truncated, a consequence of the acrodine cap departure. An enamel layer seems to have been eroded on MPV 2022.1.23.7, on the two bigger teeth.

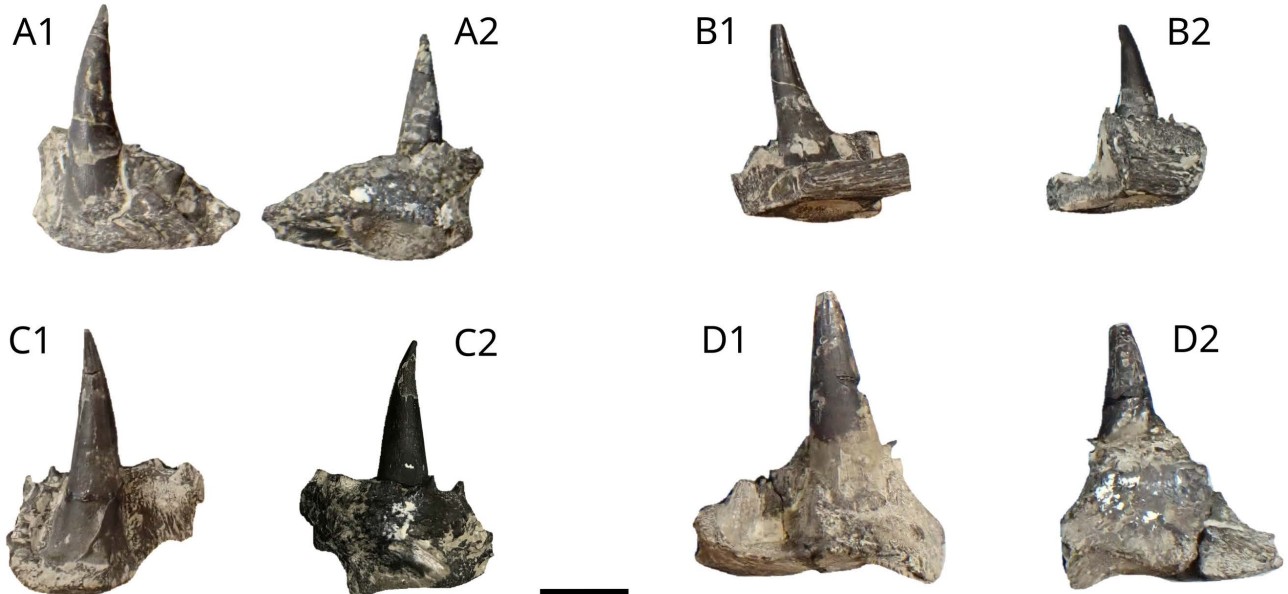

**Fig 20. *Orthocormus* cf. *tenuirostris* coronoids fragments MPV 2022.1.23.2 (A) to 2022.1.23.5 (D).** Photograph in lingual view (1) and labial view (2). Scale bar: 5 mm.

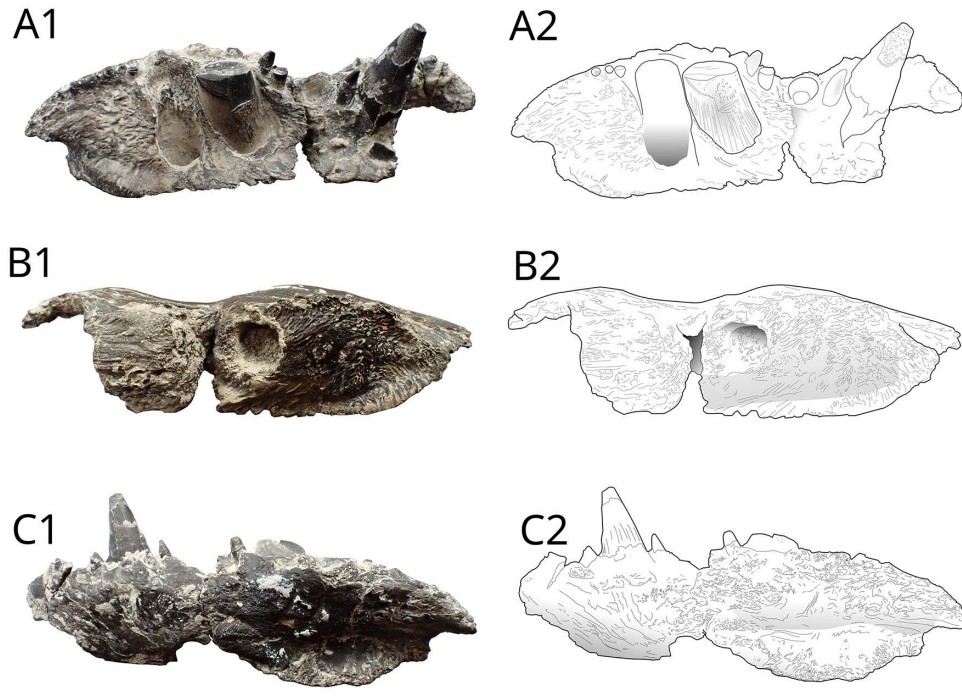

**Fig 21. *Orthocormus* cf. *tenuirostris* coronoid fragment MPV 2022.1.23.7:** Photograph (left) and drawing (right) in (probably left?) labial view (A), ventral view (B) and lateral view (C). Scale bar: 20 mm.

Actinopterygii indet.

## Material

A batch of 31 vertebrae MPV 2022.1.13 (Pennetier collection); A batch of 33 vertebrae MPV 2014.2.183 (Charles collection); A batch of 47 vertebrae V176R (Ranson collection).

## Vertebrae

Small amphicoelous vertebrae (concave anteriorly and posteriorly) are regularly found in Vaches Noires Cliffs. Three batches are housed in the Paléospace collection: V176R, MPV 2014.2.183 and MPV 2022.1.13. The first batch V176R contains 47 vertebrae, the second MPV 2014.2.183 contain 33 vertebrae and the third MPV 2022.1.13 contains 31 vertebrae. The centra are well ossified, lack a marked mid-length constriction, and range in maximum diameter from 2 to 20 mm. Their overall morphology is smooth, with no prominent grooves.

We illustrate the general morphology of the centra using vertebrae from batch 2022.1.13 (Fig 22). The first centrum is shown in axial view (Fig 22A1), lateral view (Fig 22A2), and in either ventral or dorsal view (Fig 22A3). This type of morphology is typical for centra measuring 5 mm or less, characterized by a very smooth surface.

The other centra (Fig 22B–E) exhibit variations in shape and coloration and are generally less smooth, although the overall morphology remains consistent. The figured centrum (Fig 22F), shown in lateral view, is partially broken in the presumed ventral region (lower part of the specimen). However, the upper part clearly displays the general shape seen in most of the specimens.

One vertebra of particular interest is described in this section. It originates from the V176R batch and measures a maximum of 15 mm in height and 8.5 mm in width (Fig 23). The bases of the neural arches and parapophyses are preserved, with the latter serving as rib attachment sites. The centrum is marked by deep grooves on both sides. This morphology is reminiscent with most actinopterygians, who possess well-ossified vertebra, with strongly attached parapophysis. But the overall morphologies cannot allow further identification, with a lot of vertebrae eroded, lacking any diagnostic characteristics to decide between chondrichthyans and actinopterygians. For some of the vertebrae with deep lateral grooves, we cannot exclude Synechodontiformes Duffin and Ward, 1993 (Neoselachii Compagno, 1977), who possess those types of centra [76]. Further analysis involving thin section might be the way to differentiate between cartilaginous and bony structures, but this is not the goal of this paper.

Sarcopterygii Romer, 1955

Actinistia Cope, 1871

Mawsoniidae Schultze, 1993

*Trachymetopon* Hennig, 1951

*Trachymetopon* sp.

## Material

Three pterygoids IGR 153441–153442 (Bara collection) and MPV 2023.2.3 (Paleospace collections); One palatoquadrate IGR 153443 (Gendry collection); Three basisphenoids IGR 153444, IGR 153445 (Bara – Gendry collection); Basisphenoid sutured to a partial parasphenoid MPV 2024.1.1 (Daumont collection).

## Remarks

Based on Cavin et al. [24], all small coelacanths bones are attributed to young specimens of *Trachymetopon* for the following reasons: the basisphenoids present a notochord pit ventrally visible between the two processus connectens and

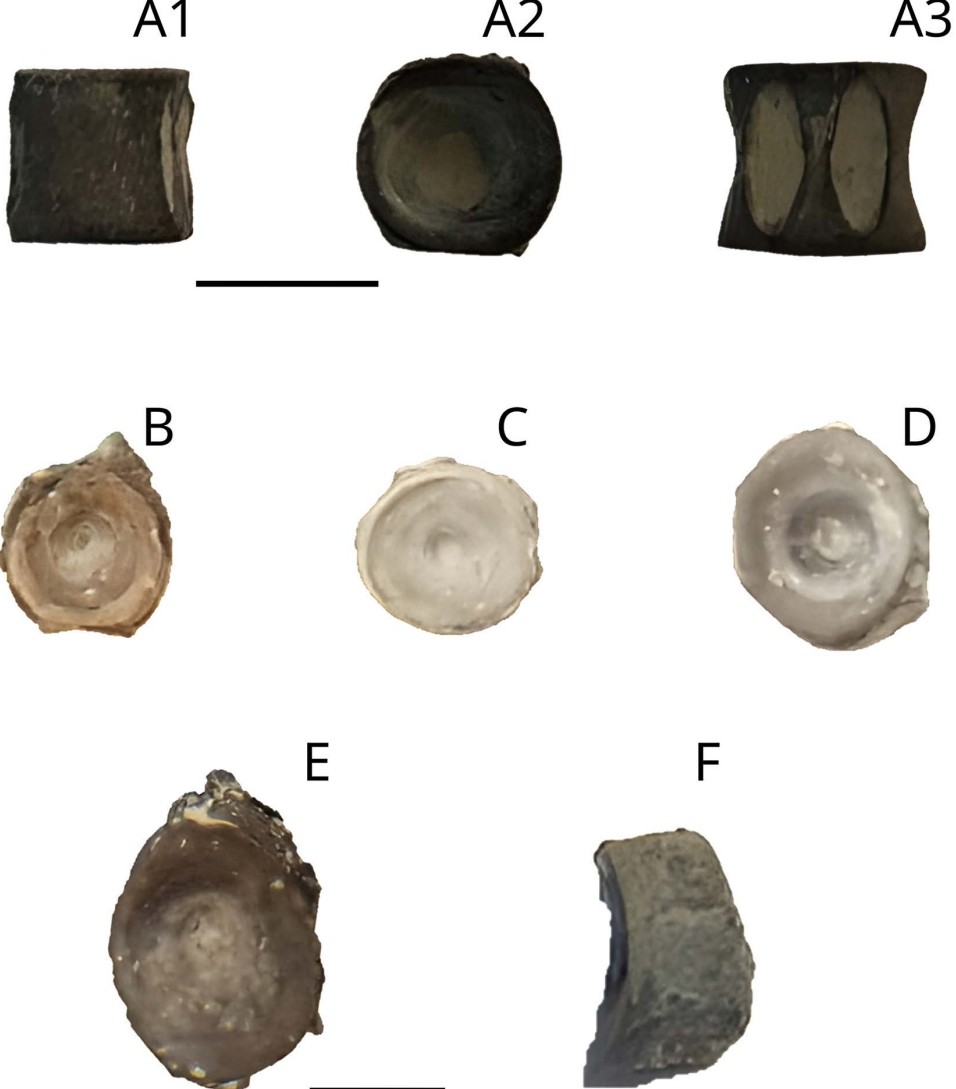

**Fig 22. Actinopterygii indet. representative vertebra centra from batch 2022.1.13 (Pennetier collection).** Scale bar: 5 mm for A1–A3; 10 mm for B–F.

an inner surface carved by ridges or cavities. Remains tend to have very coarse ridges, probably trabecular bones, visible due to weathering.

## Pterygoids

The three pterygoids are from the left side, and only the posterior part corresponding to the strongest ossified part of theses bones is preserved (). All pterygoids present an acute angle between the posteroventral and the anteroventral edge (average: 57°) compared to larger specimens of *Trachymetopon*, such as MPV 2012.1.1, SMNS.95889 [23] and MJML K785 [77], in which this angle is much more opened (≈ 86°).

On IGR 153441–153442 (Fig 24) and MPV 2023.2.3 (Fig 26) a posterior ridge originates just dorsally to the quadrate insertion surface. This ridge delimits dorsally the posterior margin of the metapterygoid. Another ridge located anteriorly

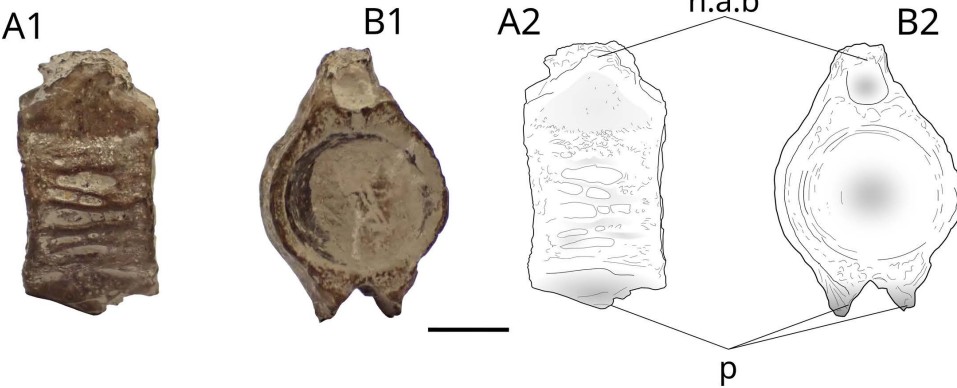

**Fig 23. Actinopterygii indet. from Bacth V176R. Photograph (A1, B1) and drawing (A2, B2) of a Teleostei indet.** vertebra from batch V176R. **n.a.b**, neural arch basis; **p**, parapophysis. Scale bar: 5 mm.

from the first delimits a pit. The latter is the insertion point for the jaw adductor muscles in coelacanths [78]. The bone surface of IGR 153441–153442 presents ridges. On IGR 153442 (Fig 24C and D) and MPV 2023.2.3 (Fig 26) a small surface bearing infra-millimetrical teeth is present on the lingual side.

## Basisphenoids

The basisphenoid is a strongly ossified bone that is therefore easily preserved. In coelacanths, it forms the anterior element of the intracranial joint between the ethmosphenoid (anterior) and the otoccipital (posterior) parts. Specimens IGR 153444 and IGR 153445 (Figs 27 and 28) have well-developed sphenoidal condyles. They are triangular in IGR 153444 (Fig 27) and rather rounded in IGR 153445 (Fig 28). The condyles are separated from each other by a wide notch comparable to the one present in the large specimen MNH GEPI V5778 [24]. The antotic processes expand dorsolaterally with part of the sutural surface with the posterior parietal preserved. The left antotic process is preserved in IGR 153444 while the right one is preserved in IGR 153445.

The dorsum sellae forms a low wall that ventrally limits the opening of the cranial cavity. The processus connectens are elongated with parallel margins, and curve slightly antero-ventrally in their central part. They reach the ventral side of the bone where the parasphenoid was sutured, visible in specimen IGR 153444 (Fig 27). MPV 2024.1.1 (Fig 29) is the most complete of all specimens in which all previously described characters are visible, plus a part of the parasphenoid. A groove connected to the pituitary fossa runs along the dorsal part of the parasphenoid. On its ventral part, small teeth are present on the preserved part of the bone. The left lateral ascending wing is partially preserved but the part that connected to the lateral ethmoid is destroyed.

## Diets and trophic web

In this section, we separately analyze each taxon of bony fish recorded in the Callovian assemblage from the Vaches Noires cliffs to provide them with an estimate of their trophic level. This analysis aims to understand the ecological interactions between fishes and the diverse fauna of the Callovian sea of the Vaches Noires cliffs. The summary of fishes found in the Vaches Noires cliffs is provided in Table 2, with trophic values and occupied niches presented in Fig 30, and the trophic web reconstruction in Fig 31.

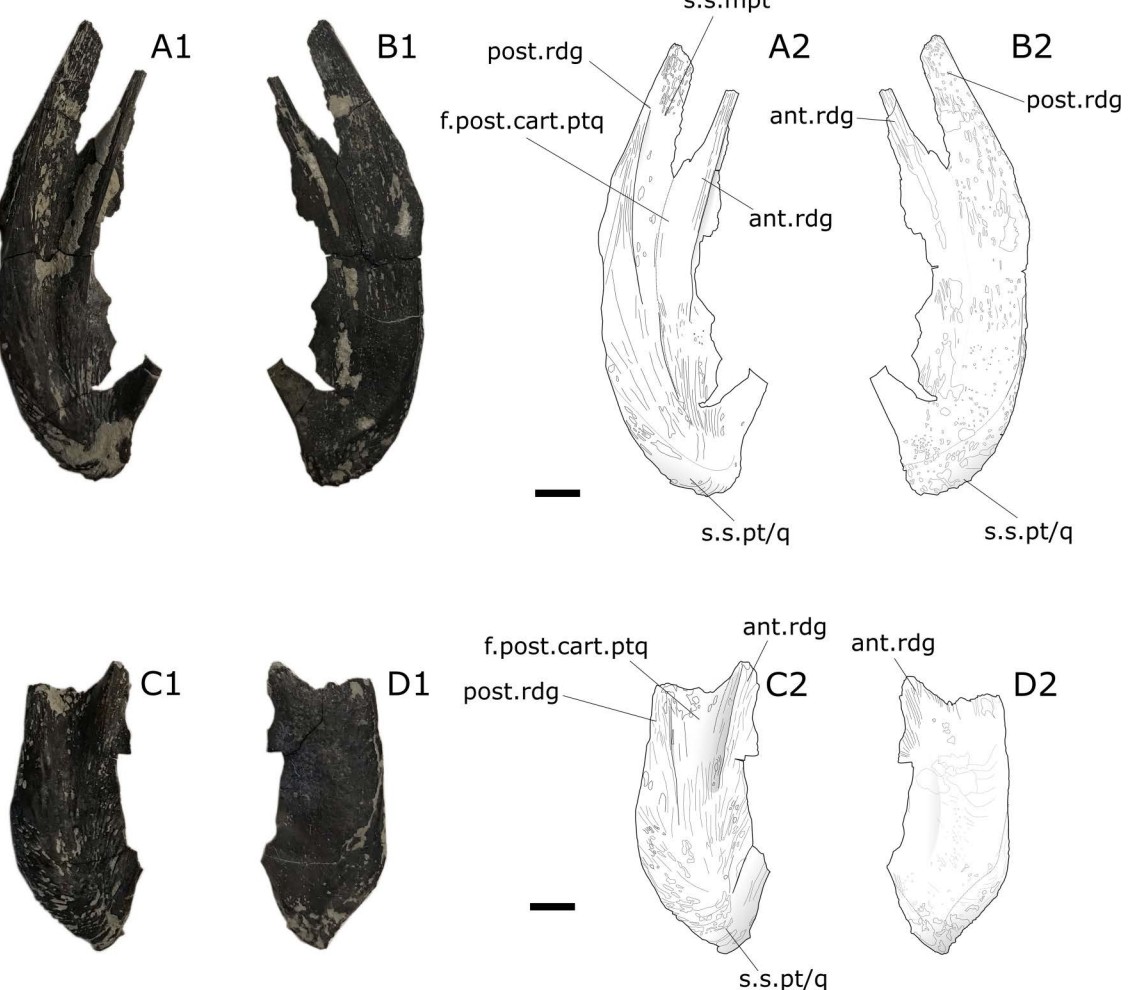

**Fig 24. *Trachymetopon* sp. IGR 153441 and IGR 153442.** Photographs (left) and drawings (right) of coelacanth pterygoid IGR 153441 in left lateral view (A1, A2), in labial view (B1, B2), and pterygoid IGR 153442 in left lateral view (C1, C2), in labial view (D1, D2). **ant.rdg**, anterior ridge; **f.post.part. ptq**, fossa for the insertion of the posterior cartilage plate of the palatoquadrate; **post.rdg**, posterior ridge; **s.s.pt/q**, suture surface of the pterygoid with the quadrate; **s.s.mpt**, suture surface of the metapterygoid. Scale bar: 5 mm.

## Halecomorphs

Halecomorphs are very scarce among the Vaches Noires cliffs. One specimen of '*Eurypoma' grande* is known from the Nicolet collection. Based on the morphology of the jaws (Fig 2), we hypothesize that *'Eurypoma'* was a generalist predator, feeding on small fishes, crustaceans, molluscs, carrions, and other food opportunities. It was likely preyed upon by mesopredators such as *Cryptoclidus*, thalattosuchians and pliosaurs. We suggest a trophic level of 3.8 based on the extant bowfin, *Amia calva* (Amiiformes).

## Lepidotidae

Lepidotidae are widespread in the Callovian of the British Oxford Clay Formation. Martill [79] associated the crushing teeth structure of '*Lepidote*s' *macrocheirus* to five arched incisions present on a damaged peristome of the ammonite *Kosmoceras* cf. *obductum*. This crescent bite-mark also corresponds to the arrangement of the teeth of specimen MPV

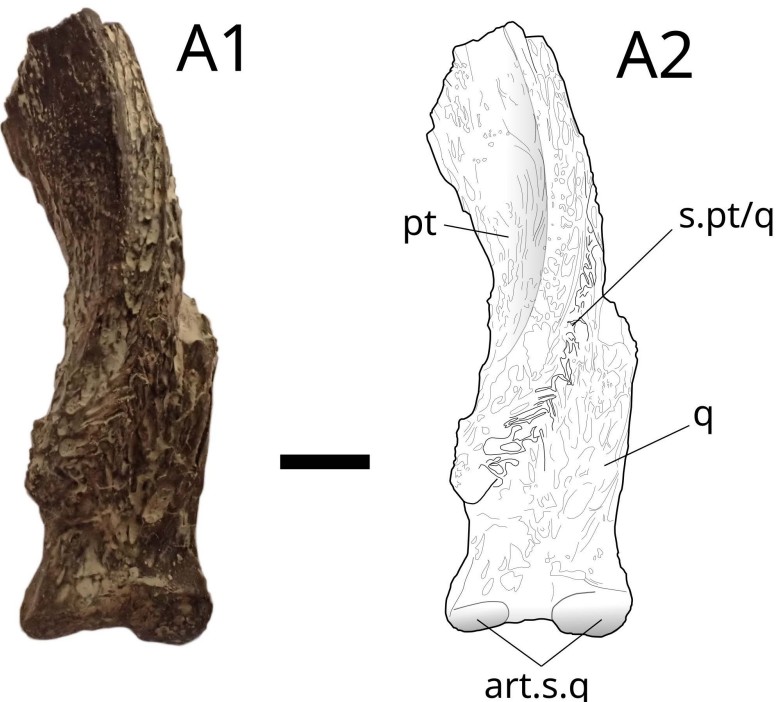

**Fig 25. *Trachymetopon* sp. IGR 153443.** Palatoquadrate IGR 153443; photograph (A1) and drawing (A2) of the palatoquadrate in posterior view. **art.s.q**, articular surface of the quadrate; **pt**, pterygoid; **q**, Quadrate; **s.pt/q**, suture of the pterygoid with the quadrate. Scale bar: 5 mm.

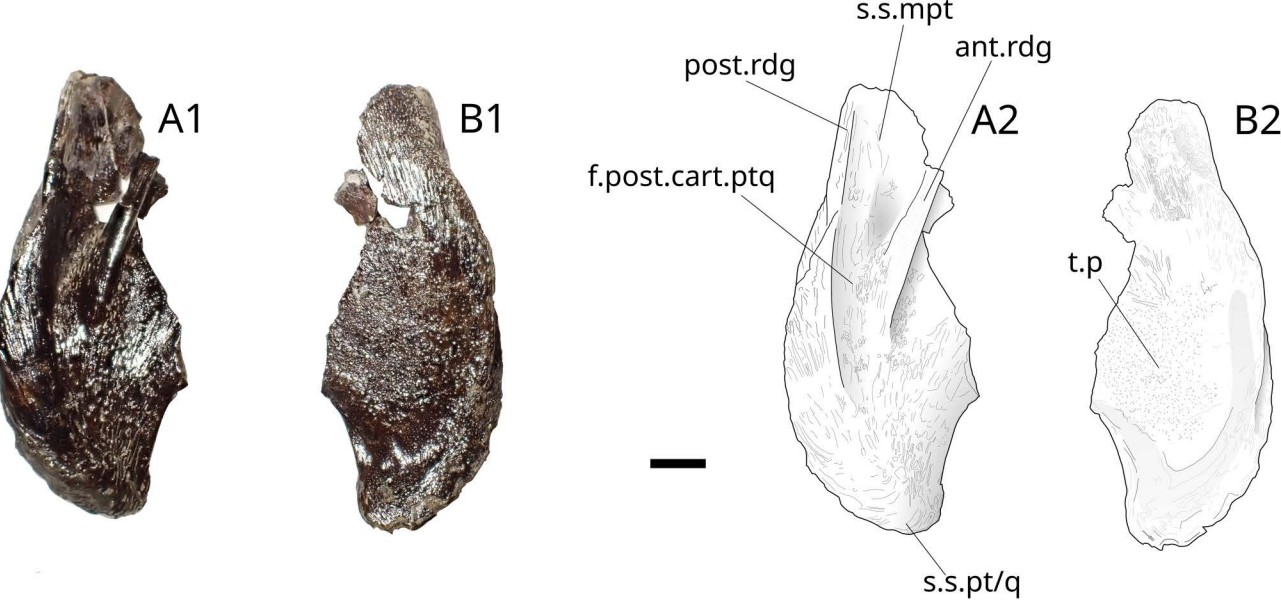

**Fig 26. *Trachymetopon* sp. MPV 2023.2.3.** Pterygoid MPV 2023.2.3in left lateral view (A1, A2) and labial view (B1, B2) **ant.rdg**, anterior ridge; **f.post.part.ptq**, fossa for the insertion of the posterior cartilage plate of the palatoquadrate; **post.rdg**, posterior ridge; **s.s.pt/q**, suture surface of the pterygoid with the quadrate; **s.s.mpt**, suture surface of the metapterygoid; **t.p**, tooth plate. Scale bar: 5 mm.

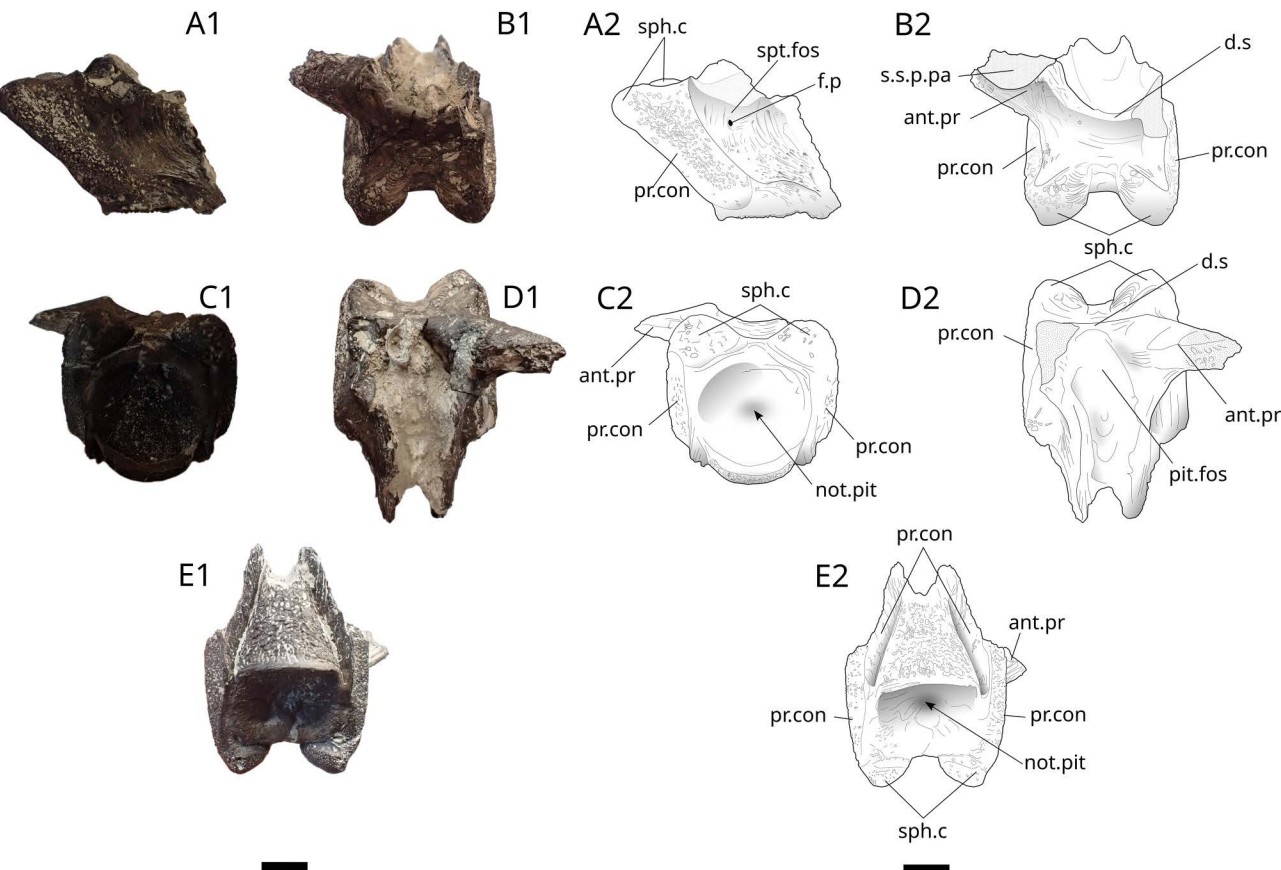

**Fig 27. *Trachymetopon* sp. IGR 153444.** Basisphenoid IGR 153444: photograph (left) and drawing (right) of basisphenoïd in left lateral view (A1, A2), dorsal view (B1, B2), postero-ventral view (C1, C2), antero-dorsal view (D1) and ventral view (E1, E2). **ant.pr**, antotic process; **d.s**, dorsum sellae; **f.p**, foramen for the profundus; **not.pit**, notochord pit; **pit.fos**, pituitary fossa; **pr.con**, processus connectens; **sph.c**, sphenoid condyles; **s.s.p.pa**, suture surface with post parietal, **spt.foss**, suprapterygoid fossa. Scale bar: 5 mm.

2022.1.15. of Lepidotidae indet. The body size of MPV 2022.1.15. was estimated using ganoid scales MPV 2022.1.16 (Fig 4) combined with the scales and body size ratio used by Cavin et al. (2015) [38] on *Adrianaichthys pankowskii*. The proportionality rule indicates a total length of about 75 cm, a body size in accordance with the size of the hyomandibula (MPV 2022.1.14, Fig 3).

The jaw fragment (MPV 2022.1.22, Fig 5) bears teeth capped by acrodine caps typical of the genus *Scheenstia*. These fishes likely fed on ammonites, benthic crustaceans, and small fishes (small-sized actinopterygians). We suggest that they were essentially consumed by mesopredators and potentially by durophagous hybodontiform sharks, able to crush massive scales and/or thick bones. We do not know where they lived in the water column, but their robust body and massive skull probably indicate a more demersal than pelagic living environment. We assign a trophic level of 4.1 based on the modern gray triggerfish, *Balistes capriscus* (Tetraodontiformes).

## Pycnodontiformes

Two distinct pycnodonts have been recognized in the Vaches Noires cliffs, the Pycnodontiformes *incertae sedis Athrodon* [21] and the Mesturidae *Mesturus* (24). Kriwet [21] figured an *Athrodon* sp. prearticular, but other occurrences are

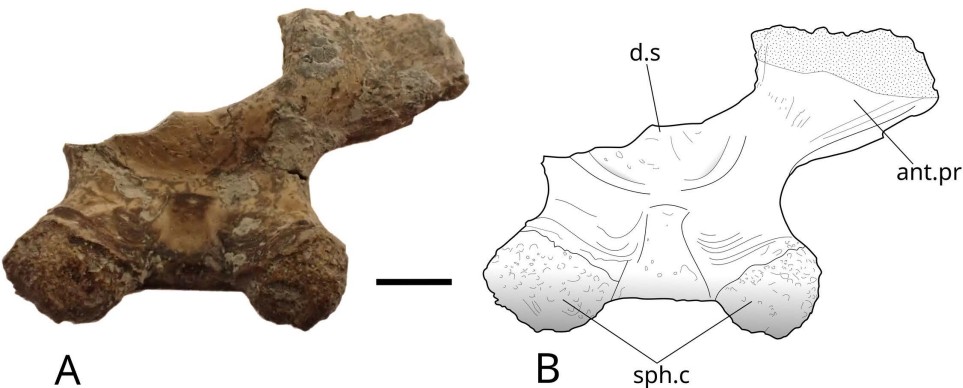

**Fig 28. *Trachymetopon* sp. IGR 153445.** Basisphenoid IGR 153445: photograph (A) and drawing (B) of basisphenoid in posterior view. **ant.pr**, antotic process; **d.s**, dorsum sellae; **sph.c**, sphenoid condyles. Scale bar: 5 mm.

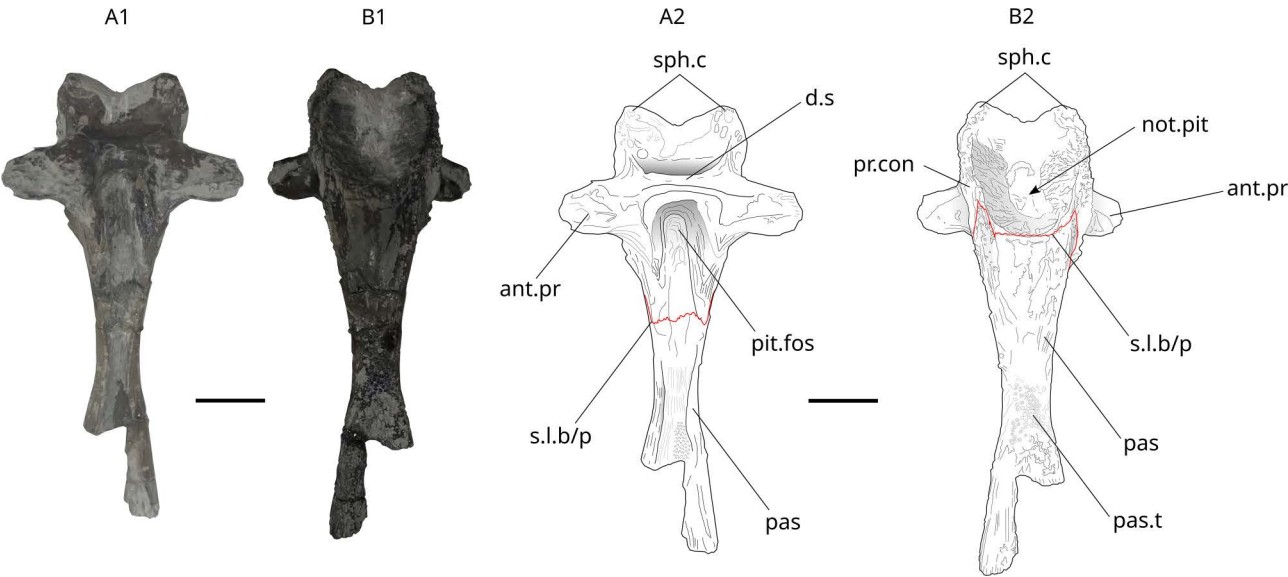

**Fig 29. *Trachymetopon* sp. mpv 2024.1.1.** Basisphenoid MPV 2024.1.1: photograph (left) and drawing (right) of basisphenoid in antero-dorsal view (A1, A2), postero-ventral (B1, B2). **ant.pr**, antotic process; **d.s**, dorsum sellae; **not.pit**, notochord pit; **pas:** Parasphenoid**; pas.t:** parasphenoidal teeth; **pit.fos**, pituitary fossa; **s.l.b/p**, suture line between basisphenoid ans parasphenoid; **pr.con**, processus connectens; **sph.c**, sphenoid condyles. Scale bar: 5 mm.

regularly found in the Callovian of Vaches Noires by fossil collectors. We follow Wenz (1967) [22] who referred a partial braincase to the genus *Mesturus* (Fig 7) and a dermal bone fragment (Fig 8). This genus is restricted to the Callovian and Late Jurassic of Central and Northern Europe [51] and is regularly found in the Oxford Clay. Similar to Lepidotidae, pycnodonts are believed to have lived relatively close to the seafloor, exhibiting a demersal lifestyle. More complete specimens of *Athrodon* and *Mesturus* have demonstrated a generalist way of swimming [80,81], not so closely related to manoeuvring swimming [82,83]. As stated by Kriwet [84], we suggest a monotypy of prey, likely gastropods and other molluscs. We assign a trophic level of 3.8 based on the living tripletail wrasse, *Cheilinus trilobatus* (Labriformes).

**Table 2. Summary table of bony fish found in the Vaches Noires cliffs. F: Frequent; LF: Locally frequent; O: Occasional; VR: Very rare.**

| | Trophic levels | Life habits | Diet | Abundance |
|---|---|---|---|---|
| *Trachymetopon* | 4,4 (Based on *Latimeria chalumnae*) | Pelagic? | Piscivorous | LF |
| *Leedsichthys* | 3,6 (Based on *Rhincodon typus*) | Pelagic | Filter-feeder | O |
| *Hypsocormus* | 4,4 (Based on *Thunnus albacares*) | Pelagic | Piscivorous | O |
| Lepidotidae indet. | 4,1 (Based on *Balistes capristus*) | Nektobenthic | Durophagous | F |
| 'Eurypoma' | 3,8 (Based on *Amia calva*) | Pelagic | Piscivorous | VR |
| *Mesturus* | 3,8 (Based on *Cheilinus trilobatus*) | Nektobenthic | Durophagous | F |
| Actinopterygii | 3,1 (Based on *Sardina pilchardus*) | Pelagic | Filter-feeder | F |

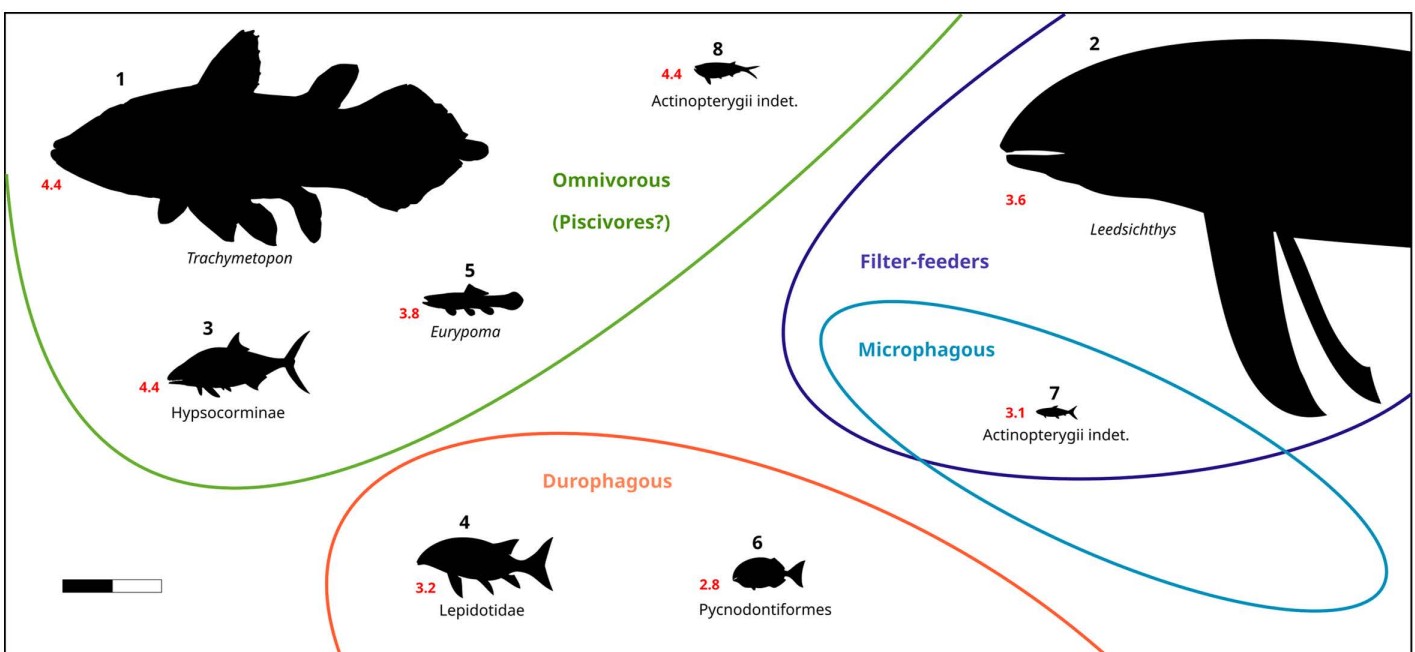

**Fig 30. Schematic representation of the estimated trophic values (red numbers) and feeding habits (green, blue, light blue and orange) of the taxa studied.** 1, *Trachymetopon* sp.; 2, *Leedsichthys problematicus*; 3, Hypsocorminae; 4, Lepidotidae; 5, Halecomorphes; 6, *Mesturus* sp.; 7, Actinopterygians indet.; 8, Actinopterygians indet. Scale bar: 1 m.

## Asthenocorminae

The giant planktivorous ray-finned fish *Leedsichthys* could reach considerable lengths (16.5 metres maximum estimated) [28,65]. *L. problematicus* exploited an important resource located at the base of the food chain, the primary consumers. It could probably also feed on small fishes, as fin whales (*Balaenoptera physalus*), humpback whales (*Megaptera novaeangliae*) and whale sharks (*Rhincodon typus*) do today [85–87]. We assign for *L. problematicus* a trophic level of 3.6 identified for the living whale shark, *Rhincodon typus* (Orectolobiformes). We do not use *Cetorhinus maximus* since is common length is 700 cm with a maximum length of 1,590 cm [37], while *R. typus* common length is 1,000 cm with maximum length circa 2,000 cm [37], closer to *Leedsichthys* average size estimation (over 1,000 cm).

*Leedsichthys* as food source: The Callovian sea was rich in mesopredators from medium to large-sized (sauropterygians, toothed pachycormids, thalattosuchians). Evidence of attacks on *L. problematicus* was recorded on dorsal fin rays

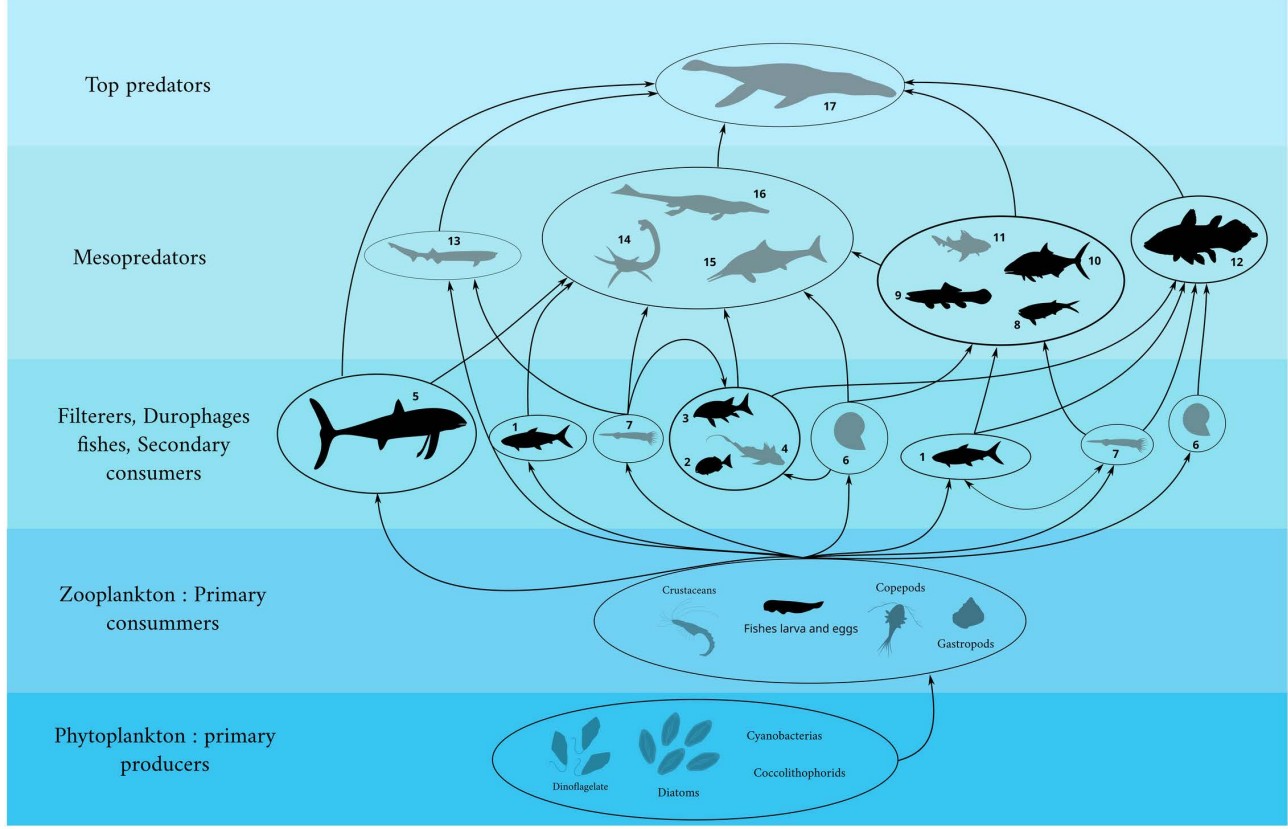

**Fig 31. Trophic web reconstruction of the Callovian sea including bony fishes and principals taxa related in the food web (not at scale).** Fishes are in black, other taxa are in grey. Trophic relationships are reconstructed and inferred from bibliography, direct evidence and by actualism. This figure is a simplified model to understand the global ecological implication of the bony fish fauna among the Callovian sea of the Vaches Noires cliffs. 1, Actinopterygii indet.; 2, Pycnodontiformes; 3, Lepidotidae; 4, Chimaeriformes; 5, *Leedsichthys*; 6, Ammonoidea; 7, Teuthoidea; 8, Actinopterygians indet.; 9, Halecomorphi; 10, Hypsocorminae; 11, Hybodontiformes; 12, *Trachymetopon*; 13, *Notidanoides*; 14, *Cryptoclidus*; 15, *Ophthalmosaurus*; 16, *Metriorhynchus*; 17, *Liopleurodon*.

(NHMUK PV P.6924) and on a caudal fin (NHMUK PV P.10000A) [65]. Injured bones have formed calluses, probably following an attack by a large pliosaur like *Liopleurodon ferox* [28]. The old or sick individuals of *L. problematicus* were probably to serve as food source to many taxa, such as thalattosuchians and sauropterygians. Scavenging evidence is known from specimen FBS 2012.4.67.30 with stomachal content of a thalattosuchian containing *L. problematicus* gill rakers [35]. Evidence of scavenging by *Metriorhynchus* on plesiosaur carcass has been documented, showing that they were opportunistic [88]. Moreover, the specimen MPV 2013.1.3 is preserved in association with a *Notidanoides muensteri* tooth, suggesting scavenging behaviour from many taxa on *L. problematicus* carcasses by pelagic fauna and on the bottom of the sea. After the mobile scavenging stage, carcasses may have served as deadfalls to provide nutritional enrichment to the sea floor, with some large fishes later becoming temporary reef environments prior to final burial (for instance on a piece of a large *Trachymetopon* braincase found in the Vaches Noires [24] and skeletons of marine reptiles from the Oxford Clay (*Ophthalmosaurus*) [11]). These taphonomic observations associated with *L. problematicus* fossils are reminiscent of what we observe today on whales falls [89,90].

## Hypsocorminae

Based on their general morphology and buccal apparatus, these pachycormids likely fed on fishes and crustaceans [10]. Sizes can be estimated by comparison with *Orthocormus cornutus* (holotype SenkM 1863) and *Hypsocormus insignis* (holotype SNSB-BSPG AS VI 4a) rostrodermethmoids, known by articulated skeletons from the Solnhofen Archipelago [68,73]. For MPV 2019.1.1, a standard size ranging between 100 cm and 150 cm is a reasonable estimate. However, since this estimation is based on isolated fragments, it should be considered only speculative. They are medium-sized predatory fishes that consumed small to medium-sized teleosteans, they were probably preyed upon by sharks, marine reptiles and other large predators. We assign a trophic level of 4.4 identified for the living yellowfin tuna, *Thunnus albacares* (Scombriformes).

## Actinopterygians indet

Leptolepidae and Pholidophoridae are present in the Peterborough member [11] but there is no conclusive evidence of their occurrence in the Vaches Noires cliffs. However, the abundance of small, smoothly ossified vertebrae in the French assemblage suggests that small-sized fishes inhabited the Callovian sea of the Vaches Noires. These remains may represent small actinopterygians, represented by taxa such as Leptolepidae or Pholidophoridae, but current evidence is insufficient to assign them to a specific group. These fishes likely occupied lower trophic levels in the vertebrate food web. They would then have been microcarnivores (suction feeding), or planktonic/zooplanctivores hunting by sight [91,92] and maybe living in schools.

By a size ratio of vertebrae from V176R (Fig 23) with the specimen of *Thrissops* cf. *formosus* JME-ETT74 [91] we roughly estimate the total body length of the individual at approximately 60 cm. This size suggests relatively large predator, capable of feeding on many of the inhabitants of the Vaches Noires cliffs' sea. All vertebral centra in the assemblage are relatively large (few are less than 2 mm). This may reflect a taphonomic bias favoring the preservation of larger, well-ossified elements, while smaller, more fragile centra are more likely to be destroyed by post-mortem processes. Alternatively, it could result from a collecting bias. We also suspect that finer recovery methods, such as sediment sieving, may reveal the presence of smaller vertebrae that are currently underrepresented. Undetermined actinopterygians from the Callovian Vaches Noires vertebrate assemblage most likely formed the base of the trophic chain but were also occupying higher trophic levels with predators located in the middle level of the food chain. They might be represented either by halecomorph-like fishes as well as small teleosteomorph-like fishes. We assign a trophic level of 3.1 identified for the living sardine, *Sardina pilchardus* (Clupeiformes).

## Coelacanths

All the new material described in this study is small, either suggesting that these specimens likely represent young individuals or represent another genus. Although *Coccoderma* has been identified in the Kimmeridge Clay [93,94], the Vaches Noires cliffs remain are too incomplete to refer to the genus level. Given that *Trachymetopon* has already been identified within the Vaches Noires cliffs, we consider all the coelacanths remains from the Vaches Noires under the genus *Trachymetopon* [23,24]. Body size estimation of specimens from the Vaches Noires have been previously made based on the reconstruction of *Axelrodichthys araripensis*, with an estimate length between 4 and 5 meters for large adults [95]. Extant *Latimeria* is ovoviviparous, meaning the young develop within the mother after hatching. The body size of extant *Latimeria* juveniles born from a 1.8-meter female is approximately 30–32 cm [96]. By size ratio, juveniles from females 4–5 meters long should measure between 60 cm and 75 cm respectively. The estimated sizes of the described individuals correspond to juveniles [24] if not neonates of large *Trachymetopon* individuals reported in the Vaches Noires [23-25].

There is little direct evidence of the diet of the extant *Latimeria*, supposed to feed on prey such as fishes, small sharks, skates and cephalopods for example [97]. We classify *Trachymetopon* as a piscivorous mesopredator.

This animal has a high trophic level, with individuals up to 4–5 meters long likely feeding on small to medium size fishes (sharks, teleosteomorphs, lepidotids, pycnodonts) and cephalopods (ammonites, belemnites) [97]. We assign a trophic level of 4.4 identified for *L. chalumnae*, but this value is here considered with caution due to the differences between the living environments and body size of the compared species (although their general morphologies are quite similar on surface).

## Ecological continuity and comparison of the Vaches Noires cliffs during the Middle to Late Jurassic

Here, we compare different localities from the Early (Sinemurian and Toarcian), Middle (Callovian), and Late (Kimmeridgian) Jurassic to illustrate the ecological continuum in the succession of bony fish assemblages (Fig 32). We exclude the sites of Cerin (late Kimmeridgian) and Solnhofen (Tithonian) because they correspond to more lagoonal settings with a

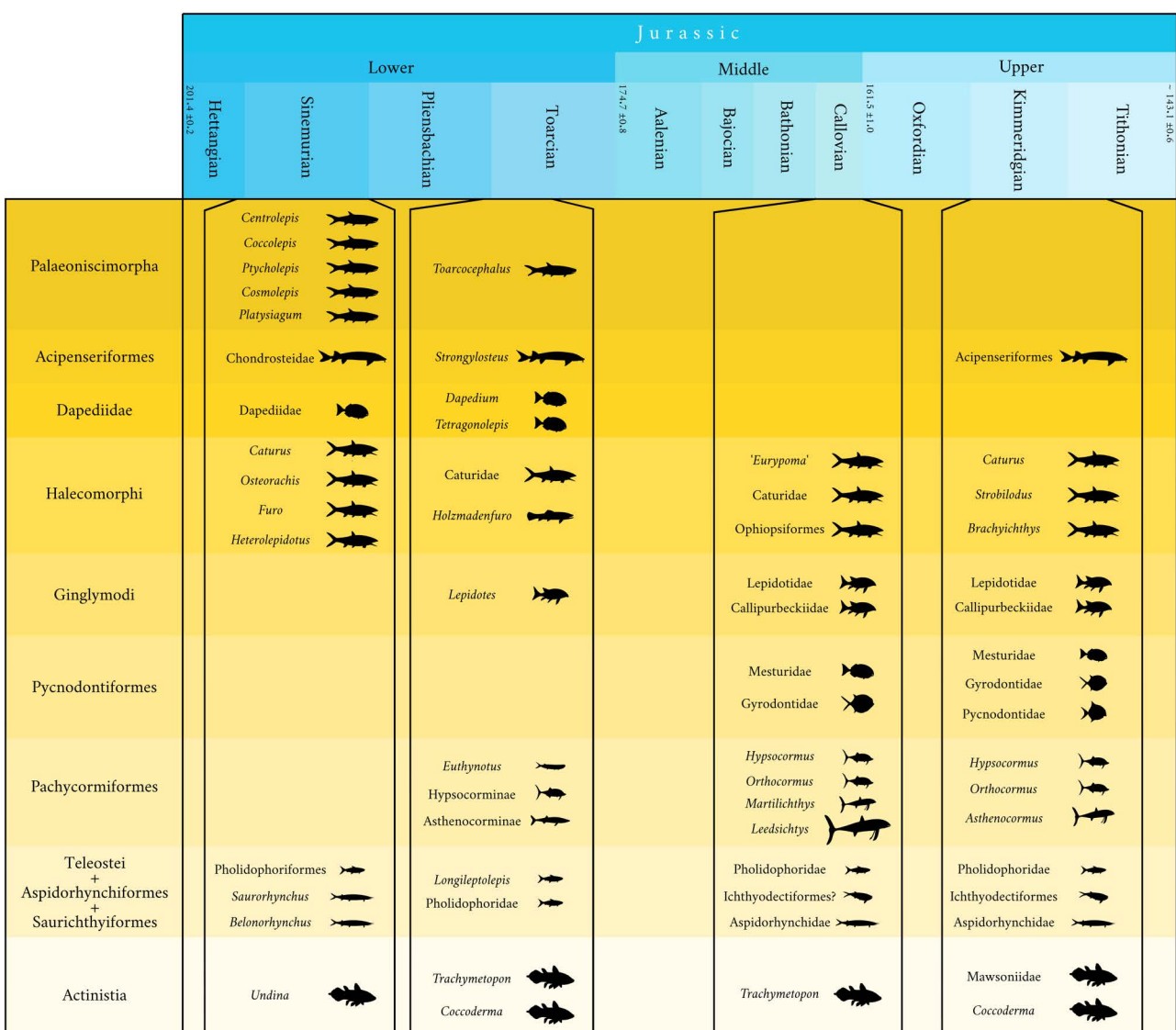

**Fig 32. Ecological 'continuum' of pelagic ichthyofauna in northwestern Europe during the Jurassic (England, Germany, France).** See text for references, list of taxa is not exhaustive.

deposit much less rich in organic matter. Instead, our objective is to examine Jurassic ecological trends by comparing similar open-pelagic environments characterized by pelletal clays (clays containing small, rounded aggregates known as pellets) [98] and rich shales (fine, laminated structures with 1–5% organic matter) found across Jurassic seas of Western Europe. Lower Lias assemblage is represented by the Hettangian and Sinemurian part of the Lyme Regis coast. Toarcian is represented by the assemblage from Holzmaden. Callovian is represented by the assemblage from the Vaches Noires cliffs and by the assemblage from the English Peterborough Member. Kimmeridgian assemblages are represented by the Kimmeridge Clay. Fig 32 is intended to show the ecological variations between Jurassic fish communities from different periods.

### Comparison between the Vaches Noires cliffs and the area of Lyme Regis (Dorset)

The Blue Lias Formation ranges from Hettangian to Sinemurian. This locality represents shallow shelf paleoenvironment, composed of dark laminated shales and limestone [99]. Paleoniscimorpha (*sensu* Lund et al., 1995) [100] represent a large part of the fauna, with *Centrolepis aspera* Egerton, 1844; *Coccolepis liassica* Woodward, 1890; *Ptycholepis* spp. Agassiz 1833*; Cosmolepis ornatus* Egerton, 1854 and *Platysiagum sclerocephalum* Egerton, 1872 [70]. Acipenceriformes are represented by *Chondrosteus acipenseroides* Egerton, 1858. Halecomorphi are represented by *Dapedium*, *Caturus*, *Osteorachis*, *Furo* and *Heterolepidotus* [70]. Unsurprisingly Pycnodontiformes are absent, but so are Lepidotidae, an unusual feature from Jurassic locality that yielded several type species [101]. Pachycormiformes are absent, but the already diverse composition of this clade in the Toarcian may suggest the existence of ghost lineages in Lower Jurassic [75]. Saurichthyiformes are represented by *Saurorhynchus* and *Belonorhynchus* [102] and Teleostei by Pholidophoriformes. The formation yielded one coelacanth genus, *Undina* [59]. This assemblage is significantly different from the Vaches Noires cliffs, with reminiscences of Paleoniscimorpha, representatives of an older ichthyofauna.

### Comparison between the Vaches Noires cliffs and Holzmaden

The Posidonienschiefer Formation, predominately quarried from the Holzmaden area near Stuttgart, is widely distributed across Baden-Württemberg. Unlike the Vaches Noires Cliffs, which corresponds to an environment close to the coast, Holzmaden represents a more open-sea environment. As a result, pycnodonts are almost completely absent from Holzmaden [103]. The Holzmaden Toarcian assemblage contains diverse halecomorphs, especially amiiforms [103], and pachycormids, with the early diverging genus *Euthynotus*, a possible Hypsocorminae "*Sauropsis",* and four early-diverging basal representatives of Asthenocorminae with *Pachycormus*, *Saurostomus*, *Germanostomus* [52,104], and *Ohmdenia* as basal representant of the filter-feeder lineage [105]. In contrast, the Vaches Noires Cliffs exhibit more derived pachycormids, such as *Hypsocormus* sp., *Orthocormus* cf. *tenuirostris*, *Leedsichthys problematicus*. A small Latimeriid coelacanth has also been described in Holzmaden by Cooper (2025) [94] and both deposits have yielded the coelacanth *Trachymetopon* [23,106]. The durophagous fishes in Holzmaden are represented by *Dapedium* [107], *Lepidotes gigas*, Agassiz 1832 and *Tetragonolepis semicincta* Bronn, 1830. The Palaeoniscimorpha *Toarcocephalus morlok* Cooper et al. 2024 is presented in Holzmaden but absent in the Vaches Noires Cliffs [108]. In Holzmaden, Acipenseriformes are represented by the genus *Strongylosteus* Jaekel 1929 [109] and the locality also yielded the ptycholepidiform, *Ptycholepis bollensis* Agassiz, 1832 [110], two species of pholidophoriforms with '*Pholidophorus*' germanicus Quenstedt, 1856 and '*Pholidophorus*' hartmanni Woodward, 1895 [111], the leptolepidiform *Leptolepis coryphaenoides* Bronn, 1830 [112] and the Saurichthyiformes *Saurorhynchus* Reis, 1892 [102].

### Comparison between the Vaches Noires cliffs and the Peterborough Member (Oxford Clay) (after [10,11])

Although the Peterborough Member and the Vaches Noires Cliffs are both Callovian in age and share similar paleoenvironmental conditions, their fish faunas display notable differences.

The Peterborough Member hosts at least one genus of pycnodont (*Mesturus* and possibly *Gyrodus*), halecomorphs such as *Caturus* and *Osteorachis*, four genera of pachycormids (*Leedsichthys*, *Hypsocormus*, *Orthocormus* and 'Sauropsis'), the aspidorhynchid *Aspidorhynchus*, and two genera of teleosteans ('*Pholidophorus*' and '*Leptolepis*'). The British locality contains several species that are absent from the Vaches Noires assemblage, including the palaeonisciform *Coccolepis*, the ginglymodian *Lepidotes*, and the dapediid *Heterostrophus*. Interestingly, no coelacanths are mentioned in the Peterborough Member, despite their presence in other localities and especially in the Vaches Noires cliffs. In contrast, the Vaches Noires Cliffs has yielded a single halecomorph, '*Eurypoma' grande* (MNHN.F.JRE45) [22,32], but *Caturus* is absent, whereas the latter is common in the Peterborough Member [10]. Furthermore, the Vaches Noires assemblage includes three genera of pachycormids (*Leedsichthys*, *Hypsocormus* and *Orthocormus*). Despite these differences, the two localities likely belonged to the same Callovian Sea, and the differences in their fish assemblages are likely due to collecting biases.

## Comparison between the Vaches Noires cliffs and the Kimmeridge Clay

The Kimmeridge Clay fish assemblage comprises Holosteans (Lepidotidae, Semionotiformes), Pycnodontiforms with three family: Mesturidae, Gyrodontidae and Pycnodontidae. In the Kimmeridge Clay, Halecomorphs are represented by the genera *Eurypoma* and by Caturidae such as *Caturus* and *Strobilodus* [113]. Ophiopsiformes are represented by *Brachyichthys* Winkler, 1862 [114]. Pachycormids are represented by three genera (*Hypsocormus*, *Orthocormus* and *Asthenocormus*). Actinistians present in the Kimmeridge Clay Formation are represented by the family Mawsoniidae (*Mawsonia* and/or *Trachymetopon*) [77] and by *Coccoderma*, the latter being absent in the Vaches Noires cliffs. The Kimmeridge Clay also contains a large indeterminate acipenceriform (Chondrostei) [115]. Teleosteomorpha is represented by *Aspidorhnychus* and *Belonostomus*, and by Teleostei [116], including the Ichthyodectiformes genera *Thrissops* and *Pachythrissops* [117].

## Discussion

The reconstruction of the bony fish assemblage of the Vaches Noires cliffs is reminiscent with that of the Peterborough Member whose food web was analysed by Martill et al. (1994) [10]. We consider both paleoecosystems to be very similar. The comparison of the Vaches Noires cliffs with four major Jurassic localities from Western Europe (Lyme Regis, Holzmaden, Peterborough Member and the Kimmeridge Clay) highlights a diversification after the Triassic-Jurassic mass extinction and a stabilization trend from the Lower (Toarcian) to the Middle Jurassic (Callovian), before the massive Upper Jurassic radiation. While there are some notable species-level differences, the broader ecological landscape remained relatively stable over time from Toarcian (Holzmaden) to Callovian (the Vaches Noires Cliffs, Peterborough Member). However, Sinemurian assemblage displays a more 'primitive' fauna, reminiscent of Triassic assemblages particularly with palaeoniscimorphs. Moreover, the variations in the abundance and diversity of certain genera across the localities (e.g., the apparent absence of coelacanths in the Peterborough Member or the limited representation of halecomorphs in the Vaches Noires Cliffs) may suggest localized adaptive radiation, driven by the availability of ecological niches or variations in environmental conditions. Differences can also be attributed to collecting bias. The comparison between the Vaches Noires Cliffs and the Kimmeridge Clay underscores a higher diversity for the latter, particularly of pycnodonts, and teleosts that represent the incipient Upper Jurassic radiation. Diversification is noticeable in pachycormids, illustrated by the split between hyspocormin and asthenocormin pachycormids during the Toarcian (with *Ohmdenia* from Ohmden). Their total range starts from the late Early Jurassic (Toarcian with *Euthynotus*) and ends at the terminal Cretaceous (Maastrichtian with *Protosphyraena* and *Bonnerichthys*) [118,119]. This makes them a very successful clade spanning almost 116 Ma.

Durophages (predominately Pycnodontiformes) are present in the Kimmeridge Clay, the Peterborough Member and the Vaches Noires cliffs. We notice a diversification among this group of fishes after the Toarcian and Middle Jurassic, particularly by the Callovian times [120], with encountered families Mesturidae, Gyrodontidae and Pycnodontidae [33]. However, Pycnodontiformes remain extremely rare throughout the Early and Middle Jurassic, particularly before the Callovian [121].

There are three, maybe four levels of piscivores in the Vaches Noires Cliffs assemblages: small-sized actinopterygians?/*Eurypoma*/*Hypsocormus*/*Trachymetopon* (*Leedsichthys* is not incorporated since its size may prevent any predatory behaviour from other fishes) and three in the Peterborough member: *Caturus*/*Hypsocormus*/'*Sauropsis*'.

The Vaches Noires cliffs assemblage contributes to our understanding of fish diversification during the Jurassic. The abundance and completeness of individual fossils and assemblages are major factors in reconstructing past biodiversity. In this respect the Vaches Noires cliffs provide valuable insights through the quality and quantity of material they have yielded. Some old and recent discoveries by amateurs suggest that fish remains can be exceptionally well-preserved in these marl deposits. Recently, a private collector (P. Branger) deposited what we believe to be a small mawsoniid neurocranium from the Callovian, discovered 45 years ago at the Vaches Noires Cliffs. The specimen has been added to the public collections of the University of Rennes and will undergo CT scanning to analyze bone topology and, hopefully, reveal new features to estimate its age, size, and life habits. An almost complete head of a '*Eurypoma*' *grande*-like fish, along with parts of its body, was discovered in 2023 on the rocky platform of the Vaches Noires cliffs, is currently in analysis. This new specimen tentatively assigned to '*Eurypoma*' *grande* is currently under study. Its remarkable state of preservation makes it a key opportunity to refine our understanding of this taxon. Ongoing preparation and analysis may provide crucial data on its anatomy, phylogenetic position, and the role of this fish within the Callovian marine ecosystem.

## Supporting information

**S1 Text. Inclusivity in global research questionnaire.**
(DOCX)

## Acknowledgments

We thank all the collectors who donated their specimens to French public institutions and shared valuable moments with us during fieldwork: Coraline Bara, Florian Daumont, Jonas Le Mort, Elisabeth and Gérard Pennetier, Patrice Rivette and Olivier Papazian. We are also grateful to Jeff Liston for insightful discussions on pachycormid remains, and to Jonas Le Mort for his helpful comments that improved this manuscript. We thank Gilles Cuny for having taken the time to look at the vertebrae to help us understand if their nature could be determined. We thank Samuel Cooper and an anonymous reviewer for their constructive comments on a previous version of this article. This work is a contribution to the project 'Evolutionary History of Mawsoniid Coelacanths' conducted at the Natural History Museum of Geneva and to the project 'Burst and Stasis in morphological evolution of Mesozoic coelacanths' funded by the Swiss National Science Foundation (https://data.snf.ch/grants/grant/207903 to L.C.).

## Author contributions

**Conceptualization:** Simon Beaufils.

**Data curation:** Simon Beaufils.

**Formal analysis:** Simon Beaufils.

**Investigation:** Simon Beaufils, Romain Vullo, Damien Gendry, Laurent Picot, Lionel Cavin.

**Supervision:** Romain Vullo, Damien Gendry, Lionel Cavin.

**Validation:** Romain Vullo, Damien Gendry, Laurent Picot, Lionel Cavin.

**Writing – original draft:** Simon Beaufils.

**Writing – review & editing:** Romain Vullo, Damien Gendry, Laurent Picot, Lionel Cavin.

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
