## [Decision Letter · Decision Letter 0]

22 Oct 2025

PONE-D-25-45591Composition and paleoecology of the mid-Jurassic bony fish assemblage of the Vaches Noires cliffs (Normandy, France).PLOS ONE

Dear Dr. Beaufils,

Thank you for submitting your manuscript to PLOS ONE. After careful consideration, we feel that it has merit but does not fully meet PLOS ONE’s publication criteria as it currently stands. Therefore, we invite you to submit a revised version of the manuscript that addresses the points raised during the review process. Your manuscript has received two detailed, constructive peer reviews. Both reviewers are in agreement that your study would provide an important contribution to the field, but also question a substantial number of the taxonomic identifications made. Both reviewers provide detailed information regarding the identifications that they question, thereby providing you with a clear rationale for re-examining the specimens in question. Considering the substantial number of taxonomic assignments that need to be re-examined, changed, or provided with a stronger rationale, I am returning the manuscript to you as a major revision. We would welcome a substantially revised version of your manuscript, which clearly has merit for helping elucidate a major mid-Jurassic bony fish assembly. Please make sure to submit a detailed response to the reviewers comments with your revised manuscript, that provides a rationale for each of the highlighted taxonomic identifications and all other points raised by the reviewers.

We look forward to receiving your revised manuscript.

Kind regards,

Leon Claessens, Ph.D.

Academic Editor

PLOS ONE

Journal Requirements:

5. Please ensure that you refer to Figure 14 in your text as, if accepted, production will need this reference to link the reader to the figure.

Reviewers' comments:

Reviewer's Responses to Questions

**Comments to the Author**

1. Is the manuscript technically sound, and do the data support the conclusions?

Reviewer #1: Yes

Reviewer #2: Partly

2. Has the statistical analysis been performed appropriately and rigorously?

Reviewer #1: N/A

Reviewer #2: N/A

3. Have the authors made all data underlying the findings in their manuscript fully available?

Reviewer #1: Yes

Reviewer #2: Yes

4. Is the manuscript presented in an intelligible fashion and written in standard English?

Reviewer #1: Yes

Reviewer #2: Yes

5. Review Comments to the Author

Reviewer #1: Dear authors,

I have thoroughly reviewed your submitted manuscript, titled “Composition and paleoecology of the Mid-Jurassic bony fish assemblage of the Vaches Noires cliffs (Normandy, France)” to the best of my knowledge on the subject. This work presents a comprehensive faunal description of the Callovian bony fish assemblage at Vaches Noires – an important but understudied Middle Jurassic marine locality with an interesting vertebrae assemblage.

I wish to begin my review by congratulating the authors for their work. The bony fish assemblage of this region has not previously been reviewed in detail, and I believe that their contribution will serve as a very useful and widely cited publication in the future. For the most part, the anatomical descriptions are well written, concise, informative and nicely illustrated. With this being said, I am not fully convinced with some of the taxonomic and anatomical identifications made by the authors. Of course, I have not examined the materials in person and are only basing my judgments on the figures provided in the authors’ manuscript, but I strongly urge the authors to take a second look at some of their specimens.

I have uploaded an annotated PDF of the manuscript with my detailed comments, most of which are grammatical/stylistic. My major concerns are as follows:

Coelacanths – It is interesting to see that coelacanth remains are present in the French Callovian, as these are one of the major Jurassic fish groups strangely absent from the English Oxford Clay Formation. Based on my recent work on coelacanth pterygoids, The specimens illustrated by the authors, in my opinion, look to be closer in morphology to Coccoderma (Laugiidae) than they do to Trachymetopon (Mawsoniidae). Please see my comments in the manuscript for further details. If the authors believe these remains to belong to a young mawsoniid, then they should explain their justification for ruling out Latimeriidae and Laugiidae as plausible candidates.

Pycnodonts – this section is fine, but see my comments on the manuscript. Also note that the specimen in Fig 25C is figured in the wrong orientation (anterior and posterior opposed relative to specimens in parts A and B). the wider medial teeth are usually always located at the posterior in pycnodontiforms vomers/prearticulars which have well organized tooth rows. Specimen in part B also appears to have some second-generation pathological teeth which have replaced a missing tooth on the left primary lateral row. The authors may wish to highlight this, but it is not all that important for the scope of the paper, more of an interesting observation which suggest it comes from an adult individual rather than a juvenile.

Pachycormidae – the authors appear to give the youngest stratigraphic range for this group as ‘Hettangian’, listing Euthynotus and Pachycormus as present in the Sinemurian. However, there are no records of Pachycormidae prior to the Toarcian. Previous records of this family in the Sinemurian derive from Agassiz (1833-1844), although all of these pre-Toarcian occurrences have since been proven as invalid or highly dubious. I am also of the opinion that the authors’ hypsocormine sample is more diverse than is reported. Certainly, the rostrodermethmoid in Fig. 16 can, in my opinion, be referred with confidence to Hypsocormus leedsi Woodward, 1889, and the anterior coronoid in Fig. 19 with two enlarged coronoids tusks be referred to Orthocormus cf. tenuirostris (Woodward, 1889) (Hypsocormus tenuirostris in older literature. The authors also use outdated informal terminologies for the subfamilies (e.g., ‘toothed clade’ and ‘suspension-feeding clade’) please use formal clade names in the manuscript (Hypsocorminae + Asthenocorminae). The authors also make reference to a ‘Leedsichthys head’ in a private collection. This specimen is both intriguing and potentially significant. Is it possible for authors to figure this specimen, perhaps in the supplementary data?

Lepidotidae – this section is sound, although I believe the specimen figured in Fig 21B-C is not a quadrate (also listed as a ceratohyal by the authors in the materials section), but instead the broken dorsal half of a hyomandibula. Please see my comments in the annotated manuscript for justification.

Teleostei – Based on the size and gross morphology of the illustrated chordocentra, and given the groups present in the Callovian from neighboring sites, leptolepids are a plausible candidate, more so than pholidophoriforms which do not have these true chordocentra.

Discussion – this section needs some work and in some areas is flawed. The taxonomic comparisons between the French Callovian and other Jurassic marine localities are interesting and ambitious, but there are many inaccuracies in the faunal assemblages given by the authors. Between the text and the comparative faunal lists provided in Fig. 32, many taxonomic groups are missing from localities where they are known to be present (e.g., Chondrosteidae, Saurichthyidae, Coccolepididae missing from Toarcian) and others misattributed, or attributed without evidence (e.g., Pachycormidae in Sinemurian; aspidorhynchids in Toarcian [perhaps conflated with saurichthyids?]). The authors do not provide a faunal description of the Sinemurian fauna (I assume based on the Lyme Regis + Charmouth Blue Lias Group?) as they do for the other localities included in Fig. 32. I have given some detailed comments on the faunal assemblages of these sites in the manuscript.

Overall, this work is highly ambitious and very important for advancing our understanding of Middle Jurassic bony fish communities, and one which certainly is worthy of publication, following revision of the points I have highlighted above. Due to number of issues, I must unfortunately recommend major revisions are needed before this manuscript is fit for publication. If the authors can satisfactorily address these concerns, then I would be happy to see this work published in PLoS ONE.

Thank you for inviting me to review this very interesting and important manuscript. I wish the authors well with the revision process and hope to see this work published in the near future. If the authors have any questions regarding my comments, then I welcome them to reach out to me via email (Samuel.cooper@smns-bw.de).

Sincerely,

Sam Cooper, Staatliches Museum für Naturkunde Stuttgart, Germany.

Reviewer #2: Dear authors

Composition and paleoecology of the mid-Jurassic bony fish assemblage of the Vaches Noires cliffs (Normandy, France) is an interesting topic. Unfortunately, little material of this locality is available, and what is there is usually so poorly preserved that it can hardly be identified.

I have inserted my comments directly into the attached PDF. Here are just some of my main points where I think the manuscript needs to be updated:

Eurypoma is not within Caturoidea (see for example López-Arbarello and Ebert 2023).

Why do you think your specimens of Fig. 25 belong to the genus Athrodon? A label or a supposed identification by S. Wenz is not sufficient.

Your “Teleostei” vertebra seem too large for Teleostei of the Middle Jurassic. The only teleost in the Middle to Upper Jurassic with completely ossified vertebra centra of this size (~5 mm) is the Ichthyodectiform Thrissops (see for example Nybelin 1964, Ebert 2025). But to me these vertebrae look completely different. To me the vertebrae in Fig 28A looks more like those in the halecomorph Amiopsis or Ionoscopus with the typical two deep depressions (see Grande & Bemis 1998).

Fig. 28B-E: For me, the figured vertebrae look more like chondrichthyan vertebrae. These vertebrae are too large for teleost vertebrae of the Jurassic. The larges Thrissops vertebrae from the Upper Jurassic (Kimmeridgian/Tithonian) are approximately 7 mm wide und 8 mm high and they were very rare. Mid-Jurassic Ichthyodectiformes were presumably smaller and all other teleost with completely ossified vertebra centra at that time were not larger than 10 to 20 cm. Thrissops vertebrae have a small outer rim that immediately descends to the centre.

Teleost are most probably present, but you should look for teleost vertebrae in the sizes <5 mm. These small vertebrae are often overlooked or simply not sampled.

Do not use the taxon name Lepidotidae (see for example López-Arbarello 2012).

Fig. 32: Lepidotidae Owen, 1860, was later added to Semionotidae by Woodward (junior synonym) and is no longer in use (see López-Arbarello 2012). Aspidorhynchiformes and Amiidae are not present in the Toarcian. Do you have any evidence for Orthogonikleithridae in the Callovium? I only know this Family from the Upper Jurassic. Ichthyodectidae are only present in the Cretaceous (see Cavin et al. 2013). The taxon name Pholidophoridae is no longer in use for the Middle and Upper Jurassic. Grimmenodus is only found in Northern Germany and not in Holzmaden.

Your Sinemurian locality in Fig. 32 is probably Lyme Regis. Here, many taxa are missing (Ophiopsiformes like Furo, Heterolepidotus, Osteorhachis; Leptolepidae…).

For Coccolepididae in Holzmaden see Cooper et al. 2024.

For Latimeriidae in Holzmaden see Cooper 2025.

The name Paraleptolepis Arratia & Thies, 2001 was preoccupied and was changed into Longileptolepis Arratia, 2003.

Ophiopsiformes are missing in all localities.

Change Dapedium into Dapediidae.

For this reason, this manuscript is not publishable in its present form and I think major revisions are needed.

I will send the PDF with additional minor corrections directly to Plosone since I cannot upload it here.

6. PLOS authors have the option to publish the peer review history of their article (what does this mean? ). If published, this will include your full peer review and any attached files.

**Do you want your identity to be public for this peer review?**  For information about this choice, including consent withdrawal, please see our Privacy Policy .

Reviewer #1: **Yes:**  Dr Samuel Cooper

Reviewer #2: No

---

## [Author Response · Author response to Decision Letter 1]

13 Jan 2026

Dear editor, Dear Reviewers

We thank both reviewers for the very complete and constructive reviews.

Supplementary comments in text have been addressed.

The main text has been entirely restructured to fit systematics, with Actinopterygians first and Sarcopterygians at the end.

Response to reviewer 1:

- Coelacanths – It is interesting to see that coelacanth remains are present in the French Callovian, as these are one of the major Jurassic fish groups strangely absent from the English Oxford Clay Formation. Based on my recent work on coelacanth pterygoids, The specimens illustrated by the authors, in my opinion, look to be closer in morphology to Coccoderma (Laugiidae) than they do to Trachymetopon (Mawsoniidae). Please see my comments in the manuscript for further details. If the authors believe these remains to belong to a young mawsoniid, then they should explain their justification for ruling out Latimeriidae and Laugiidae as plausible candidates.

Response : After discussion with Dr Cooper, we agree with him about coelacanths remain, but the very fragmentary nature of the Vaches Noires cliffs coelacanths make an identification to the genus Coccoderma very risky. Since Trachymetopon has already been found in the Vaches Noires cliffs, we prefer to keep our identification until more remains are found.

- Pycnodonts – this section is fine, but see my comments on the manuscript. Also note that the specimen in Fig 25C is figured in the wrong orientation (anterior and posterior opposed relative to specimens in parts A and B). the wider medial teeth are usually always located at the posterior in pycnodontiforms vomers/prearticulars which have well organized tooth rows.

Response : We modified the pycnodont Fig 25C in accordance with the comments (Prearticular reoriented) and it has been redescribed after discussion between authors.

- Pachycormidae – the authors appear to give the youngest stratigraphic range for this group as ‘Hettangian’, listing Euthynotus and Pachycormus as present in the Sinemurian. However, there are no records of Pachycormidae prior to the Toarcian. Previous records of this family in the Sinemurian derive from Agassiz (1833-1844), although all of these pre-Toarcian occurrences have since been proven as invalid or highly dubious. I am also of the opinion that the authors’ hypsocormine sample is more diverse than is reported. Certainly, the rostrodermethmoid in Fig. 16 can, in my opinion, be referred with confidence to Hypsocormus leedsi Woodward, 1889, and the anterior coronoid in Fig. 19 with two enlarged coronoids tusks be referred to Orthocormus cf. tenuirostris (Woodward, 1889) (Hypsocormus tenuirostris in older literature. The authors also use outdated informal terminologies for the subfamilies (e.g., ‘toothed clade’ and ‘suspension-feeding clade’) please use formal clade names in the manuscript (Hypsocorminae + Asthenocorminae). The authors also make reference to a ‘Leedsichthys head’ in a private collection. This specimen is both intriguing and potentially significant. Is it possible for authors to figure this specimen, perhaps in the supplementary data?

Response : After discussion with Dr Cooper, we agree with him about the Pachycormid issues, and we have changed some of our identification: Hypsocorminae from Sinemurian was a mistake, it has been removed. We refer the rostrodermethmoid to Hypsocormus cf. leedsi. We refer the coronoids fragments to Orthocormus cf. tenuirostris. Older terminology has been replaced by Hypsocorminae and Asthenocorminae. The Leedsichthys head will be discussed further in private with Dr Cooper, since the head is published, but it is in a private collection and is inaccessible. We removed this sentence from the text, since we haven’t heard from the specimen in many years, and it cannot be analyzed.

- Lepidotidae – this section is sound, although I believe the specimen figured in Fig 21B-C is not a quadrate (also listed as a ceratohyal by the authors in the materials section), but instead the broken dorsal half of a hyomandibula. Please see my comments in the annotated manuscript for justification.

Response : We reexamined the specimen Fig 21B-C, and we are not convinced by the Hypsocorminae identification. After careful examination, we concluded it corresponds more to a hyomandibula of Lepidotidae. It is reminiscent of Lepidotes gloriae. Since this specific fragment is very difficult to identify, we remained cautious in our determination.

- Discussion – this section needs some work and in some areas is flawed. The taxonomic comparisons between the French Callovian and other Jurassic marine localities are interesting and ambitious, but there are many inaccuracies in the faunal assemblages given by the authors. Between the text and the comparative faunal lists provided in Fig. 32, many taxonomic groups are missing from localities where they are known to be present (e.g., Chondrosteidae, Saurichthyidae, Coccolepididae missing from Toarcian) and others misattributed, or attributed without evidence (e.g., Pachycormidae in Sinemurian; aspidorhynchids in Toarcian [perhaps conflated with saurichthyids?]). The authors do not provide a faunal description of the Sinemurian fauna (I assume based on the Lyme Regis + Charmouth Blue Lias Group?) as they do for the other localities included in Fig. 32. I have given some detailed comments on the faunal assemblages of these sites in the manuscript.

Response : The discussion section and Fig 32 have been completely revised. We now include Lyme Regis locality, with a dedicated section in the text. All missing taxonomic identification have been added.

To reviewer 2:

- Eurypoma is not within Caturoidea (see for example López-Arbarello and Ebert 2023)

Response : Eurypoma phylogeny placement has been corrected

- Why do you think your specimens of Fig. 25 belong to the genus Athrodon? A label or a supposed identification by S. Wenz is not sufficient.

Response : Pycnodont vomers are neither Gyrodus nor Mesturus due to uncrenulated nature of the teeths. Eomesodon and Proscinetes have ellipsoid teeth, where our specimens possess rounded teeth strikingly similar to Athrodon wittei and to the prearticular founded in the Marnes de Dives, both figured by Kriwet, 2008.

- Do not use the taxon name Lepidotidae (see for example López-Arbarello 2012).

Response : Lepidotidae is a valid taxon since Lopez-Arbarello and Wenckler, 2016 (Neopterygian phylogeny the merger assay)

- Aspidorhynchiformes and Amiidae are not present in the Toarcian.

Response : Aspidorhynchiformes and Amiidae in Toarcian has been removed

- Do you have any evidence for Orthogonikleithridae in the Callovium? I only know this Family from the Upper Jurassic

+ Ichthyodectidae are only present in the Cretaceous (see Cavin et al. 2013)

+ The taxon name Pholidophoridae is no longer in use for the Middle and Upper Jurassic. Grimmenodus is only found in Northern Germany and not in Holzmaden.

+ The name Paraleptolepis Arratia & Thies, 2001 was preoccupied and was changed into Longileptolepis Arratia, 2003.

Response : All concerns in Fig 32 have been addressed (Paraleptolepis into Longileptolepis, Orthogonikleithridae in Callovian and all the mentioned above).

- Your Sinemurian locality in Fig. 32 is probably Lyme Regis. Here, many taxa are missing (Ophiopsiformes like Furo, Heterolepidotus, Osteorhachis; Leptolepidae…).

Response : The Sinemurian locality has been added (Lyme Regis) with a description in the main text.

- Ophiopsiformes are missing in all localities + For Latimeriidae in Holzmaden see Cooper 2025 + For Coccolepididae in Holzmaden see Cooper et al. 2024 + Dapedium into Dapediidae

Response : The same comments have been made by the first reviewer for Coccolepididae and Latimeriidae in Holzmaden. These taxa have been added, just as Ophiopsiformes in all localities and Halecomorphi genera such as Heterolepidotus or Osteorachis.

For Both reviewers:

- From reviewer 1: Teleostei – Based on the size and gross morphology of the illustrated chordocentra and given the groups present in the Callovian from neighboring sites, leptolepids are a plausible candidate, more so than pholidophoriforms which do not have these true chordocentra.

- From reviewer 2: Your “Teleostei” vertebra seem too large for Teleostei of the Middle Jurassic. The only teleost in the Middle to Upper Jurassic with completely ossified vertebra centra of this size (~5 mm) is the Ichthyodectiform Thrissops (see for example Nybelin 1964, Ebert 2025). But to me these vertebrae look completely different. To me the vertebrae in Fig 28A looks more like those in the halecomorph Amiopsis or Ionoscopus with the typical two deep depressions (see Grande & Bemis 1998).

Fig. 28B-E: For me, the figured vertebrae look more like chondrichthyan vertebrae. These vertebrae are too large for teleost vertebrae of Jurassic. The larges Thrissops vertebrae from the Upper Jurassic (Kimmeridgian/Tithonian) are approximately 7 mm wide und 8 mm high and they were very rare. Mid-Jurassic Ichthyodectiformes were presumably smaller and all other teleost with completely ossified vertebra centra at that time were not larger than 10 to 20 cm. Thrissops vertebrae have a small outer rim that immediately descends to the centre.

Teleosts are most probably present, but you should look for teleost vertebrae in the sizes <5 mm. These small vertebrae are often overlooked or simply not sampled.

Response : Comments in the text about syntax have been considered and modified. Both comments about ‘Teleostei’ vertebrae are well founded and we thank the reviewers for the constructive review. Since both reviews were diverging on the identification, we asked another opinion from Gilles Cuny, expert in chondrichthyans. It seems that without thin section, it is practically impossible to determine whether the vertebrae are Chondrichthyans or Actinopterygians, but some of them, especially the one in Fig 29 is identified as Actinopterygian. We agree on the Amiopsis similarity, comments in text have been made on this point. We have agreed to identify most vertebrae as undetermined actinopterygians. We think this is not the core information of the article, and some vertebrae are small, indicating small fishes, the latter being eaten by larger fishes the same way if they were teleostean, halecomorphs or any other kind of actinopterygian, which is the main information to retain.

To editor:

- Fig 1 is a map remade entirely by S. Beaufils and is made by using free online and open access resources from BRGM., which have an open-science policy: https://www.brgm.fr/en/activities/knowledge-dissemination-open-science

- Following the link https://infoterre.brgm.fr/page/conditions-dutilisation-donnees you can download the ETALAB licence ouverte pdf information https://www.etalab.gouv.fr/wp-content/uploads/2017/04/ETALAB-Licence-Ouverte-v2.0.pdf regarding BRGM copyright policies, which permit reuse and derivative works with attribution.

- Fig 14 is properly cited

- Reviewers’ recommendations about literature have been followed according to relevance and to our responses

---

## [Decision Letter · Decision Letter 1]

11 Feb 2026

PONE-D-25-45591R1Composition and paleoecology of the mid-Jurassic bony fish assemblage of the Vaches Noires cliffs (Normandy, France).PLOS One

Dear Dr. Beaufils,

Thank you for submitting your revised manuscript to PLOS ONE. The revised manuscript has been re-evaluated by the same reviewers who examined the first version. We thank you for the careful amendments made to the original manuscript and find the revised document to be nearly ready for acceptance, pending some final and relatively straightforward minor corrections and clarifications. These last modifications should not take you long to implement. Below, a short list of last comments by Dr. Cooper are appended to the message. The other reviewer had only one minor comment, which I copy here: "In Fig. 32: Dapedium and Tetragonolepis are not Halecomorphi".

Please review the comments made by both reviewers and implement these last corrections. In cases where you disagree with the proposed correction, please explain your rationale. Once I have received your revised manuscript, I look forward to helping move your paper toward publication. Please submit your revised manuscript by Mar 28 2026 11:59PM. If you will need more time than this to complete your revisions, please reply to this message or contact the journal office at plosone@plos.org . Please include the following items when submitting your revised manuscript:

We look forward to receiving your revised manuscript.

Kind regards,

Leon Claessens, Ph.D.

Academic Editor

PLOS One

Journal Requirements:

Reviewers' comments:

Reviewer's Responses to Questions

**Comments to the Author**

1. If the authors have adequately addressed your comments raised in a previous round of review and you feel that this manuscript is now acceptable for publication, you may indicate that here to bypass the “Comments to the Author” section, enter your conflict of interest statement in the “Confidential to Editor” section, and submit your "Accept" recommendation.

Reviewer #1: (No Response)

2. Is the manuscript technically sound, and do the data support the conclusions?

Reviewer #1: Yes

3. Has the statistical analysis been performed appropriately and rigorously? 

Reviewer #1: N/A

4. Have the authors made all data underlying the findings in their manuscript fully available?

Reviewer #1: Yes

5. Is the manuscript presented in an intelligible fashion and written in standard English?

Reviewer #1: Yes

6. Review Comments to the Author

Reviewer #1: Dear authors and editor,

I am more than satisfied with the responses by the authors who have addressed all of my previous comments well. Again, I congratulate the authors on their work, for the formation and more importantly the age of the fossils they have presented are poorly documented in the literature on Mesozoic fishes. This study contributes an important wealth of knowledge on actinopterygian diversity and biogeography in the Middle Jurassic.

I have noticed a few minor things that the authors should address below, most are related to grammar, but nothing major. Otherwise, I would be happy to see this manuscript published in its present form, pending these very minor corrections.

sincerely,

Samuel Cooper, Stuttgart

Line 380 – do not need commas ‘,’ before word “in Normandy” as ‘in’ is a bridge word.

Line 381 – Early (spelling).

Line 522 – Orthocormus cf tenuirostris should have (Woodward, 1889) in brackets as he originally described this species as Hypsocormus tenuirostris

Line 589 – lower case ‘a’ in “actinopterygians” – change “who” to ‘that’

chondrichthyians and actinopterygians should be lower case as they are informal taxonomic names, unless at the beginning of a sentence.

594 – Do the authors mean ‘centra’ (pleural)?

594 – change “ thin section: to ‘thin sectioning’ or ‘histology’

595 – change “state” to ‘differentiate’

758 – there are several large pliosaurs in the oxford clay that could have injured the tail fin specimen, perhaps edit slightly to say “following a probable attack by a large pliosaur, like Liopleurodon ferox.”

808 – change to “Halecomorphi-like” or “halecomorph-like”

809 – change to ‘teleosteomorph-like fishes’ or just ‘teleosteomorphs’.

813 – change to either “other genera” or “another genus”.

814 – Kimmeridge Clay Formation

814 – remains (pleural) – consider changing to say “…too incomplete to confidently refer to genus level”

819 – I do not know why previous authors have scaled up these Trachymetopon specimens based on Axelrodichthys when they refer them to Trachymetopon. Trachymetopon liassicum, the type and only species is based on a virtually complete and fully articulated specimen from Ohmden, housed in the University of Tubingen.

868 – Lyme Regis is the name for one small town along the Jurassic coast, it is not called the Lyme Regis coast. Change to just ‘…area of Lyme Regis (Dorset)’.

868 – change to ‘Holzmaden area’ as this incorporates a lot of the nearby localities where most of these taxa are found – e.g., Ohmden, Bad Boll.

876 – delete comma after ‘ranges’ ‘..represents a shallow…’

879 – Ptycholepis spp. – present are Ptycholepis curta (curtus), P. monolifer, P. minor ect.

881 – specifically by Chondrosteus acipenseroides Egerton ex. Agassiz

882 – Tetragonolepis is (to my knowledge) not found in Lyme Regis – Tetragonolepis is a Toarcian genus named by Bronn (1830) which Agassiz used for describing his species that would later be moved to Dapedium Leach, including those from Lyme Regis.

884 – what do you mean by yielding more than 50 type species? 50 type species from Lyme Regis?

886 – change “at Toarcian” to “in the Toarcian”

893 – say “The Posidonienschiefer Formation, predominately quarried from the Holzmaden area near Stuttgart, is widely distributed across Baden-Württemberg.

889 – delete ‘basal’

889 – also a true hypsocormine based on an isolated mandible, Sauropsis veruinalis may represent an additional hypsocormine

891 – delete “a” after Ohmdenia (Saurostomus and Germanostomus are also transitional forms, although less derived than Ohmdenia).

904 – ‘has also been described by Cooper (2025)’

909 – Acipenseriformes

948 – consider revising as “The Kimmeridge Clay also contains a large indeterminate acipenseriform (Chondrostei)”

948 – Teleosteomorpha is represented by…

1004 – change “in analysis” to ‘under study’ and delete the following sentence, and change in the following sentence “this taxon” to “Eurypoma’ grande.

7. PLOS authors have the option to publish the peer review history of their article (what does this mean? ). If published, this will include your full peer review and any attached files.

**Do you want your identity to be public for this peer review?** For information about this choice, including consent withdrawal, please see our Privacy Policy .

Reviewer #1: **Yes:**  Samuel L. A. Cooper

---

## [Author Response · Author response to Decision Letter 2]

19 Feb 2026

Dear editor, Dear Reviewers

We thank both reviewers for the last comments

Reviewer comments are in red, our responses are in black. Supplementary comments in text have been addressed.

Last comments have all been implemented, from both reviewers.

Comments from Dr Cooper :

819 – I do not know why previous authors have scaled up these Trachymetopon specimens based on Axelrodichthys when they refer them to Trachymetopon. Trachymetopon liassicum, the type and only species is based on a virtually complete and fully articulated specimen from Ohmden, housed in the University of Tubingen.

It’s because we had a well-made reconstruction of the skeleton of A. araripensis at our disposal. Furthermore, all these coelacanths, especially Mawsoniids, have a pretty conservative morphology. We think that reconstructing size either with Trachymetopon or Axelrodichthys will give the same results.

Line 380 – do not need commas ‘,’ before word “in Normandy” as ‘in’ is a bridge word.

Line 381 – Early (spelling).

Line 522 – Orthocormus cf tenuirostris should have (Woodward, 1889) in brackets as he originally described this species as Hypsocormus tenuirostris

Line 589 – lower case ‘a’ in “actinopterygians” – change “who” to ‘that’

chondrichthyians and actinopterygians should be lower case as they are informal taxonomic names, unless at the beginning of a sentence.

594 – Do the authors mean ‘centra’ (pleural)?

594 – change “ thin section: to ‘thin sectioning’ or ‘histology’

595 – change “state” to ‘differentiate’

758 – there are several large pliosaurs in the oxford clay that could have injured the tail fin specimen, perhaps edit slightly to say “following a probable attack by a large pliosaur, like Liopleurodon ferox.”

808 – change to “Halecomorphi-like” or “halecomorph-like”

809 – change to ‘teleosteomorph-like fishes’ or just ‘teleosteomorphs’.

813 – change to either “other genera” or “another genus”.

814 – Kimmeridge Clay Formation

814 – remains (pleural) – consider changing to say “…too incomplete to confidently refer to genus level”

868 – Lyme Regis is the name for one small town along the Jurassic coast, it is not called the Lyme Regis coast. Change to just ‘…area of Lyme Regis (Dorset)’.

868 – change to ‘Holzmaden area’ as this incorporates a lot of the nearby localities where most of these taxa are found – e.g., Ohmden, Bad Boll.

876 – delete comma after ‘ranges’ ‘..represents a shallow…’

879 – Ptycholepis spp. – present are Ptycholepis curta (curtus), P. monolifer, P. minor ect.

881 – specifically by Chondrosteus acipenseroides Egerton ex. Agassiz

882 – Tetragonolepis is (to my knowledge) not found in Lyme Regis – Tetragonolepis is a Toarcian genus named by Bronn (1830) which Agassiz used for describing his species that would later be moved to Dapedium Leach, including those from Lyme Regis.

886 – change “at Toarcian” to “in the Toarcian”

893 – say “The Posidonienschiefer Formation, predominately quarried from the Holzmaden area near Stuttgart, is widely distributed across Baden-Württemberg.

889 – delete ‘basal’

889 – also a true hypsocormine based on an isolated mandible, Sauropsis veruinalis may represent an additional hypsocormine

891 – delete “a” after Ohmdenia (Saurostomus and Germanostomus are also transitional forms, although less derived than Ohmdenia).

904 – ‘has also been described by Cooper (2025)’

909 – Acipenseriformes

948 – consider revising as “The Kimmeridge Clay also contains a large indeterminate acipenseriform (Chondrostei)”

948 – Teleosteomorpha is represented by…

1004 – change “in analysis” to ‘under study’ and delete the following sentence, and change in the following sentence “this taxon” to “Eurypoma’ grande.

All these comments above about grammar have been taken into account and changed in both manuscript and manuscript with track changes

884 – what do you mean by yielding more than 50 type species? 50 type species from Lyme Regis?

Yes. In Dineley and Metcalf, 1999, it is said, I quote “Lyme Regis is the type locality for 50 or more species of fish.”

Comments from Reviewer 2 :

In Fig. 32: Dapedium and Tetragonolepis are not Halecomorphi

Fig 32 has been modified in accordance with this comment. Dapedium and Tetragonolepis have been moved to a line of their own, the Dapediidae. The figure has been downloaded and is replacing the old Fig 32.

---

## [Editor Report · Decision Letter 2]

22 Feb 2026

Composition and paleoecology of the mid-Jurassic bony fish assemblage of the Vaches Noires cliffs (Normandy, France).

PONE-D-25-45591R2

Dear Dr. Beaufils,

We’re pleased to inform you that your manuscript has been judged scientifically suitable for publication and will be formally accepted for publication once it meets all outstanding technical requirements.

Kind regards,

Leon Claessens, Ph.D.

Academic Editor

PLOS One
---

## [Editor Report · Acceptance letter]

PONE-D-25-45591R2

PLOS One

Dear Dr. Beaufils,

I'm pleased to inform you that your manuscript has been deemed suitable for publication in PLOS One. Congratulations! Your manuscript is now being handed over to our production team.

Kind regards,

on behalf of

Dr. Leon Claessens

Academic Editor

PLOS One